# A Correction Algorithm for Rotor-Induced Airflow and Flight Attitude Changes during Three-Dimensional Wind Speed Measurements Made from A Rotary Unmanned Aerial Vehicle

Yanrong Yang[1+], Yuheng Zhang[1+], Tianran Han[1], Conghui Xie[2], Yayong Liu[2], Yufei Huang[1], Jietao Zhou[1], Haijiong Sun[1], Delong Zhao[3], Kui Zhang[4], Shao-Meng Li[1*]

[1]College of Environmental Sciences and Engineering, Peking University, Beijing 100871, China

[2]Laboratory of Gas Instrument Testing, Center for Environmental Metrology, National Institute of Metrology, Beijing 100029, China

[3]Beijing Weather Modification Center, Beijing 100089, China

[4]Beijing Wisdominc Technology Co., Ltd, Beijing 100070, China

+Contributed equally to the work

*Correspondence to: Shao-Meng Li (shaomeng.li@pku.edu.cn)*

**Abstract.** A hexacopter unmanned aerial vehicle (UAV) was fitted with a three-dimensional sonic anemometer to measure three-dimensional wind speed. To obtain accurate results for three-dimensional wind speeds, we developed an algorithm to correct biases caused by the rotor-induced airflow disturbance, UVA movement, and attitude changes in the three-dimensional wind measurements. The wind measurement platform was built based on a custom-designed integration kit that couples seamlessly to the UAV, equipped with a payload and the sonic anemometer. Based on an accurate digital model of the integrated UAV-payload-anemometer platform, computational fluid dynamics (CFD) simulations were performed to quantify the wind speed disturbances caused by the rotation of the UAV's rotor on the anemometer during the UAV's steady flight under headwind, tailwind, and crosswind conditions. Through analysis of the simulated data, regression equations were developed to predict the wind speed disturbance, and the correction algorithm for rotor disturbances, motions, and attitude changes was developed. To validate the correction algorithm, we conducted a comparison study in which the integrated UAV flew around a meteorological tower on which three-dimensional wind measurements were made at multiple altitudes. The comparison between the corrected UAV wind data and those from the meteorological tower demonstrated an excellent agreement. The corrections result in significant reductions in wind speed bias caused mostly by the rotors, along with notable changes in the dominant wind direction and wind speed in the original data. The algorithm enables reliable and accurate wind

speed measurements in the atmospheric boundary layer made from rotorcraft UAVs.

**Keywords**: UAV; Rotor Disturbance; Three-dimensional Wind; Correction Algorithm

**1 Introduction**

Wind measurement is crucial in various fields of research and application, including meteorology and environmental sciences. Accurate wind characteristics facilitate modeling of atmospheric transport patterns (Gryning et al., 1987; Stockie, 2011), remote sensing data verification (Drob et al., 2015), model input data assimilation (Gousseau et al., 2011; Vardoulakis et al., 2003) and digital modeling result optimization (Booij et al., 1999; Van Hooff and Blocken, 2010). In particular, wind profile measurements near surface can improve the understanding of atmospheric boundary layer (ABL) dynamics and micrometeorological turbulence at the surface (Seibert et al., 2000), allowing detailed understandings and model description of energy and mass exchanges between air and surfaces and transport processes.

The recent development of unmanned aerial vehicles (UAVs) has provided an opportunity for the measurement of wind fields in three dimensions with high spatial resolutions (Mcgonigle et al., 2008; Martin et al., 2011; Kim and Kwon, 2019). The small size, low flight altitude, high mobility and ability to assemble sensing devices make UAVs ideal platforms from which to measure wind in the ABL (Thielicke et al., 2021; Shaw et al., 2021; Stewart et al., 2021). Multirotor UAVs allow flexible control of flight attitude and stationary hovering, and can carry varying payloads depending on the number of rotors (Villa et al., 2016; Riddell, 2014; Bonin et al., 2013; Stewart et al., 2021), offering significant advantages in capturing high-resolution wind characteristics in low-altitude conditions (Anderson and Gaston, 2013; Mcgonigle et al., 2008).

UAVs are often employed to measure wind characteristics both directly and indirectly. Indirect measurement methods involve utilizing pre-installed sensors on the UAV (Elston et al., 2015), in conjunction with specialized flight patterns and wind retrieval algorithm (Bonin et al., 2013; Rautenberg et al., 2018; Gonzalez-Rocha et al., 2019) to achieve wind speed measurement. While these methods offer advantages of operational simplicity and cost-effectiveness, their core principle relies on inversely estimating wind speed through dynamic parameters such as thrust, attitude angles, and flight velocity (Crowe et al., 2020; Donnell et al., 2018; Sikkel et al., 2016; Simma et al., 2020). However, their accuracy is critically dependent on both the measurement precision of inertial measurement unit (IMU)

and the computational reliability of inversion algorithms. Specifically, inherent noise interference in IMU sensors (e.g., gyroscope's angular rates can be severely affected by external disturbances up to 0.5 °/s) (Hoang et al., 2021; Neumann and Bartholmai, 2015), combined with uncertainties in parameter configuration within inversion algorithms (the root mean squared errors (RMSE) of wind speed estimation is 1-1.4) (Bonin et al., 2013), can lead to significant deviations in wind speed estimations. Furthermore, these methods typically assume constant aerodynamic parameters for UAVs, an assumption that often fails to hold in practical complex wind field environments (Bonin et al., 2013).

In contrast, direct measurement methods entail installing additional wind sensors on the UAV to obtain real-time wind information in the field. Porous probes (Soddell et al., 2004; Spiess et al., 2007), pitot tubes (Niedzielski et al., 2017; Langelaan et al., 2011), and anemometers (Rogers and Finn, 2013; Nolan et al., 2018) are commonly used sensors. Sonic anemometers are a more prevalent choice for rotorcraft UAVs, capable of measuring wind speed by detecting changes in the speed of sound travel between different sensors (Thielicke et al., 2021). Recent experiments have demonstrated that under highly turbulent conditions, UAV equipped with properly installed sonic anemometers in wind tunnels can achieve wind speed measurements with RMSE ranging from 4.3% to 15.5% compared to bistatic lidar (Thielicke et al., 2020). Due to the increasing use of rotorcraft UAVs for wind measurements, sonic anemometers are recognized as one of the most promising methods in terms of measurement accuracy and precision.

Sonic anemometers have been mounted onto rotary-wing UAVs for measuring wind speed to varying degrees of success. Typically, an anemometer is mounted at a position along the central axis above the UAV, with data adjusted for the additional wind speed signals induced by UAV motion and attitude changes. Nevertheless, the strong airflow perturbations caused by the rotating propellers can distort real wind flow patterns and significantly affect the accuracy of wind measurements (De Divitiis, 2003). However, these distortions were not considered in the adjustment algorithms. To address this issue, researchers have developed several new correction methods. The first method involves mounting the anemometer along the central axis high above the UAV where the rotor wash effects are believed to be limited on the wind speed measurement (Shimura et al., 2018; Barbieri et al., 2019). Johansen concluded that anemometers at about 40 to 45 mm above the multi-rotor plane of small UAV the flow influences from rotors are negligible (Johansen et al., 2015). However, it may not be suitable for

hexacopters and octocopters due to the high position required, which may raise safety and flight control
concerns. The second method involves new corrections based on experiments in an indoor area to
measure wind velocity signal bias caused by the rotors during flight and then subtracting the bias
(Palomaki et al., 2017). Palomaki et al. (2017) quantified rotor-induced wind speed errors as 0.5 m/s
compared to tower-mounted anemometers and subtracted these errors from the directly measured wind
speed values in subsequent analyses (Palomaki et al., 2017). However, this method is limited by the size
of the indoor area, inadequate for full simulations of real UAV rotor speed and attitude changes during
flight, and insufficient for the development of a comprehensive correction scheme. Additionally, it does
not take into account the detailed coupling of true winds with propeller downwash. The third method is
similar to the second except the use of wind tunnels to establish a more accurate relationship between
increased air speed and UAV motion or attitude parameters (Thielicke et al., 2021; Neumann and
Bartholmai, 2015). While effective in determining numerical relationships, the method is limited by the
high cost of wind tunnel experiments (Dao et al., 2023), and more importantly, by the additional errors
introduced by reflected airflows from the wind tunnel walls and ground (Haleem, 2021; Pettersson and
Rizzi, 2008), as well as the same issues of full simulations of real UAV rotor speed and attitude changes
during flight.
The flaws in these correction methods could be addressed by using computational fluid dynamics
(CFD) simulations to analyze the airflow generated by the UAV's propellers. As far as we know, CFD
has been employed to analyze airflow patterns around drones but hasn't been utilized to correct wind
measurements obtained from UAVs (Oktay and Eraslan, 2020; Hedworth et al., 2022). In this paper, we
introduce a three-dimensional wind speed correction algorithm for sonic anemometer wind
measurements taken from a rotary UAV. This algorithm considers the rotor-induced airflow of the UAV,
based on CFD simulations, along with the UAV's motion and attitude changes during flight. The
accuracy of the algorithm is confirmed by comparing the corrected wind speeds with those measured
from a meteorological tower at multiple altitudes. These results could contribute to ongoing efforts
aimed at enhancing the performance and reliability of UAV-based wind speed measurement techniques.
Additionally, they pave the way for potential applications, such as quantifying pollutant emissions from
industrial complexes (Han et al., 2023).
**2 Method**
**2.1 Digital Model Representation and Simulation Tool**
**2.1.1 Digital Model Representation**
A six-rotor UAV (KWT-X6L-15, ALLTECH, China), equipped with six 32 cm diameter propellers
driven by M10 KV100 brushless DC motors, was the platform from which wind was measured. The
UAV has a symmetrical motor wheelbase of 1765 mm with an unloaded takeoff weight of 22.5 kg and
a maximum flight speed of 18 m s$^{-1}$. It has a flight endurance > 30 min while carrying its maximum
payload of 15 kg.
A miniature three-dimensional ultrasonic anemometer (Trisonica-Mini Wind and Weather Sensor,
Anemoment, America) allowed the measurement of wind speed under 15 m s$^{-1}$ with an accuracy of $\pm$
0.1 m s$^{-1}$ and a resolution of 0.1 m s$^{-1}$, and wind direction of 0 - 360° with an accuracy of $\pm$ 0.1° and a
resolution of 0.1°. It was set at 70 cm above the plane of the propellers of the UAV, mounted on a custom-
design carbon fiber tube and frame which was further mounted onto a rectangular carbon fiber support
base attached to the underbelly of the UAV body, to minimize the effect of propellers-induced flow on
the anemometer measurement. The $x_t$-$y_t$-$z_t$ coordinate axes of the anemometer, with its center as the
origin, were set to be parallel to the $x$-$y$-$z$ axes of the aircraft body frame. The mounting of the three-
dimensional anemometer is shown in Fig. 1(a).
A base digital model of the UAV was provided by its manufacturer for the present CFD simulations.
The digital model was further augmented with the accurate digital representation of the three-
dimensional anemometer and its mounting frame. Furthermore, considering that the UAV wind
measurements are usually tied to other air measurement applications, necessitating additional payload
attached to the UAV underbelly simultaneously. Such a payload on the UAV needs also to be included
in the digital model for the CFD simulation. In the present case, we added the digital model of a 6.37 kg
air sampler developed in our group (Yang et al., 2024) to the UAV base digital model (Fig. 1(b)).

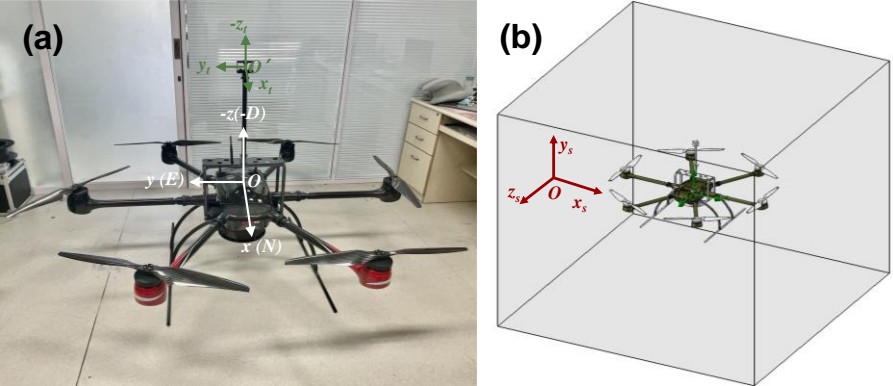


**Figure 1: The establishment of the coordinate system and numerical simulation model for the UAV wind**
**measurement platform. (a) The UAV wind speed measurement platform. (b) The digital model of the UAV**
**wind measurement platform in the 3D CFD model simulation domain.**
**2.1.2 Simulation Tool**
The CFD simulations were conducted using SolidWorks Flow Simulation 2022, a pressure-based
finite volume solver employing a fully coupled turbulence modeling approach. It employs an adaptive
Cartesian mesh approach for three-dimensional solid meshing, with the governing equations being the
Navier-Stokes equations for simulating the interaction of fluids, and the turbulence model utilizing the
standard k-ε two-equation model (Jonuskaite, 2017).
The selection of SolidWorks Flow Simulation was driven by its seamless integration with CAD
geometries, which eliminated potential errors associated with STL file conversions for our complex
multi-rotor UAV design. Additionally, its wall functions for boundary layers effectively resolve gradient
variations in boundary layers around rotating blades, reducing trial and error related to near-wall settings.
The built-in solver convergence adopts a phased approach to multiple variant scenarios, decreasing the
need for re-runs caused by insufficient convergence and thereby conserving computational costs. Its
unique turbulence model automatically determines flow regimes (laminar, transitional, and turbulent),
ensuring shorter turbulence model setup times while maintaining enhanced model accuracy (Azmi et al.,
2017; Ramya et al., 2015).
While ANSYS Fluent offers advanced transient turbulence models (e.g., DES/LES), its
computational cost for equivalent spatial resolution was typically higher than SolidWorks (Afaq and
Ahmad, 2023). Given our need to simulate over 100 operational scenarios, SolidWorks' balance of
engineering accuracy and computational tractability was deemed optimal for deriving empirical
correction algorithm.

      For CFD simulations, the complete digital model for the UAV and its payloads was set in the $x_s$-$y_s$-

$z_s$ simulation coordinate system in Solidworks, on a one-to-one scale (Fig. 1(b)).
**2.2 Simulation Scenarios**
For the UAV flight simulation, we considered over a hundred flight envelope scenarios, including
parameters such as UAV flight altitude, wind direction, and wind speed. Since the UAV's predominant
flights are within the atmospheric boundary layer, characterized by significant variability in wind speed
and directions, a flight envelope for the UAV in the simulated environments was setup for the complete
UAV digital model for flight altitudes of 30 and 1000 meters, respectively. The lower height (30 m)
represents the typical operational altitude for industrial UAV applications within the boundary layer,
while the upper height (1000 m) corresponds to the altitude where atmospheric flow transitions to more
stable, low-density free-stream conditions. These flight envelopes were designed for the UAV to subject
to headwind, tailwind, and crosswind relative to its flight direction. Under the constraint that the UAV
can only operate under true wind speeds $\leq 18$ m s$^{-1}$, and assuming the applicability of the correction
algorithm to most flight scenarios, CFD simulations were conducted for the UAV under these three wind
directions. The simulations encompassed the following flight envelopes as listed in Table 1: the UAV
flew at ground speeds of 18, 14, 10, and 8 m s$^{-1}$, respectively, and adapted to wind speeds of 1.5, 3.3,
5.4, 7.9, 10.7, and 14 m s$^{-1}$. It should be noted that the numerical simulations were conducted by
converting wind speed and ground speed into airspeed through vector synthesis.
**Table 1: The simulation flight envelope scenarios for the UAV-based wind measurement platform.**

| Wind Type | Ground Speed (m s$^{-1}$) | Wind speed (m s$^{-1}$) | Wind Type | Ground Speed (m s$^{-1}$) | Wind speed (m s$^{-1}$) | Wind Type | Ground Speed (m s$^{-1}$) | Wind speed (m s$^{-1}$) |
|---|---|---|---|---|---|---|---|---|
|  |  | 1.5 |  |  | 1.5 |  |  | 1.5 |
|  |  | 3.3 |  |  | 3.3 |  |  | 3.3 |
|  | 8 | 5.4 |  | 8 | 5.4 |  | 8 | 5.4 |
|  |  | 7.9 |  |  |  |  |  | 7.9 |
|  |  | 10.7 |  |  |  |  |  | 10.7 |
| Tailwind |  |  | Headwind |  |  | Crosswind |  | 14 |
|  |  | 1.5 |  |  | 1.5 |  |  | 1.5 |
|  |  | 3.3 |  |  | 3.3 |  |  | 3.3 |
|  | 10 | 5.4 |  | 10 | 5.4 |  | 10 | 5.4 |
|  |  | 7.9 |  |  | 7.9 |  |  | 7.9 |
|  |  | 10.7 |  |  |  |  |  | 10.7 |

| | | | | | |
|---|---|---|---|---|---|
| | | | | | 14 |
| 14 | 1.5 | 14 | 1.5 | 14 | 1.5 |
| | 3.3 | | 3.3 | | 3.3 |
| | 5.4 | | 5.4 | | 5.4 |
| | 7.9 | | 7.9 | | 7.9 |
| | 10.7 | | 10.7 | | 10.7 |
| | | | | | 14 |
| 18 | 1.5 | 18 | 1.5 | 18 | 1.5 |
| | 3.3 | | 3.3 | | 3.3 |
| | 5.4 | | 5.4 | | 5.4 |
| | 7.9 | | 7.9 | | 7.9 |
| | 10.7 | | 10.7 | | 10.7 |
| | | | 14 | | 14 |

## 2.3 Flight Parameters

The movements of the UAV through air, including takeoff, ascent/descent, attitude changes, turning, and horizontal flights, are driven by the rotary propellers, whose power requirement is closely tied to the weights of the UAV and its payload as well as the relative motions of the UAV in air. During a normal flight, the UAV adjusts its inclination angle and propeller speeds in order to achieve a set ground speed for flight. By analyzing the gravity $G$, pull $T$ and wind resistance $D$ experienced by the UAV under flight conditions (Fig. 2), its inclination angle $\theta$ and propeller rotation speed $M$ can be calculated according to Eqs. (1) - (5) (Quan, 2017).

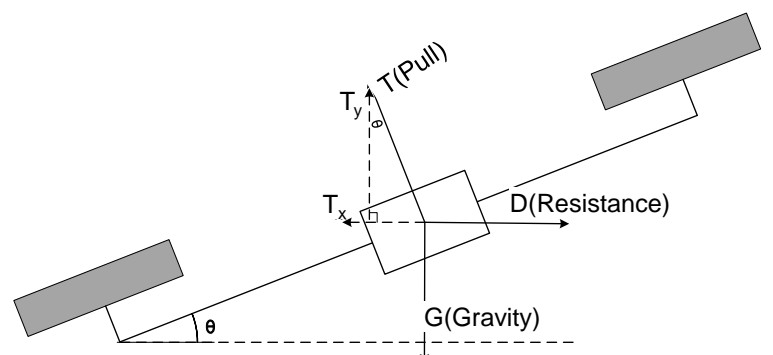

**Figure 2: Schematic diagram of forces acting on a UAV.**

$$\tan\theta \times mg = D, \tag{1}$$

$$p \times (\sin\theta \times S_{xoy} + \cos\theta \times S_{xoz}) = D, \tag{2}$$

$$0.5\rho(V_{wind} + V_{UAV})^2 = p, \tag{3}$$

$$\cos\theta \times mg = T, \tag{4}$$

$$T = C_T \times \rho \times \left(\frac{M}{60}\right)^2 \times D_p^{\ 4}, \tag{5}$$

where $\theta$ is the inclination angle of the UAV; $m$ is the combined weight of the UAV and the payloads (i.e, the air sampler and the anemometer along with its installation frame in the present case), calculated to be 28.869 kg; $g$ is the gravitational constant at 9.8 m s$^{-2}$; $D$ is the wind resistance in Newtons; $V_{wind}$ is the wind speed in m s$^{-1}$; $V_{UAV}$ is the ground speed of the UAV in m s$^{-1}$; $p$ is the wind pressure on the UAV in N/m$^2$; $S_{xoy}$ and $S_{xoz}$ are the projected surfaces of the UAV in the horizontal direction and vertical directions, determined to be 0.296 and 0.229 m$^2$, respectively; $C_T$ is the rotor pull coefficient with an experimentally determined value of 0.048542; $D_p$ is the UAV propeller diameter at 0.8128 m; $\rho$ is the air density in kg m$^{-3}$; $T$ is each rotor pull in Newton; $M$ is the rotation speed of the rotors in RPM.

The complete flight envelope was defined by combinations of critical parameters, including wind directions, wind speeds, airspeeds, ground speed, inclination, wind resistance, pull, and $M$. A series of CFD simulations were conducted to systematically evaluate the simulated wind field characteristics for each unique parameter set within this envelope.

**2.4 Simulation Parameters**

The simulation parameters primarily include the computational domain and mesh, fluid and environmental properties, as well as the rotating region. During the CFD flow simulations of the UAV using Solidworks, the computational domain dimensions (3.3 × 3.3 × 3.3 m³ ) were determined by prioritizing the analysis of flow field distribution around the anemometer while balancing computational costs. The computational domain was divided into two parts with different spatial resolutions based on the grid sizes,considering the computational time and accuracy required for resolving the details of the digital UAV model. The first part was the global domain with a grid size of 0.23 × 0.23 × 0.23 m³, providing a lower spatial resolution. The second part was a nested subdomain within the global domain, specifically defined for the position and dimensions of the anemometer to simulate the measured velocities. The grid size for this nested subdomain was set at 0.0125 × 0.0125 × 0.0125 m³, providing a higher spatial resolution. The total number of grids in the computational domain was $1.11 \times 10^8$, and the specific grid configurations are shown in Fig. 3. The wall is set as an adiabatic wall, and its roughness is set to 0. The fluid was modeled as air with characteristics of turbulent and laminar flow. To isolate the rotor-induced flow dynamics from background atmospheric turbulence, a turbulence intensity of 0.1% and a turbulence length scale of 0.012 m were set. This low turbulence intensity minimizes confounding effects from ambient atmospheric fluctuations, while the length scale corresponds to the anemometer

frame width (~0.01 m) to resolve rotor-generated eddies. These assumptions prioritize the systematic
bias correction for rotor-induced airflow. The atmospheric pressure was adjusted to $1.01 \times 10^5$ and $9.00$
$\times 10^4$ Pa at altitudes of 30 m and 1000 m, respectively, and the atmospheric temperature was assumed
to be 25 °C at both altitudes. The relative humidity at different altitudes was determined based on the
prescribed pressure and temperature corresponding to each altitude. The detailed configurations of these
parameters are listed in Table 2.
The UAV's airspeed and aerodynamic angles were configured according to the different flight
parameters described in Sect. 2.2 and 2.3. To represent the rotor digitally, six virtual cylinders of the
same volume were used to encapsulate the six rotors, with their circumferences match the rotating
trajectory of the propeller tip. These virtual cylinders were treated as the rotational regions in the CFD
simulation, with their rotation directions aligned with the actual rotation direction of the UAV's
propellers. The rotation direction from rotor No. 1 to 6 was alternately clockwise and counterclockwise,
and the rotation speed for each flight condition was obtained from Eqs. (1) - (5).

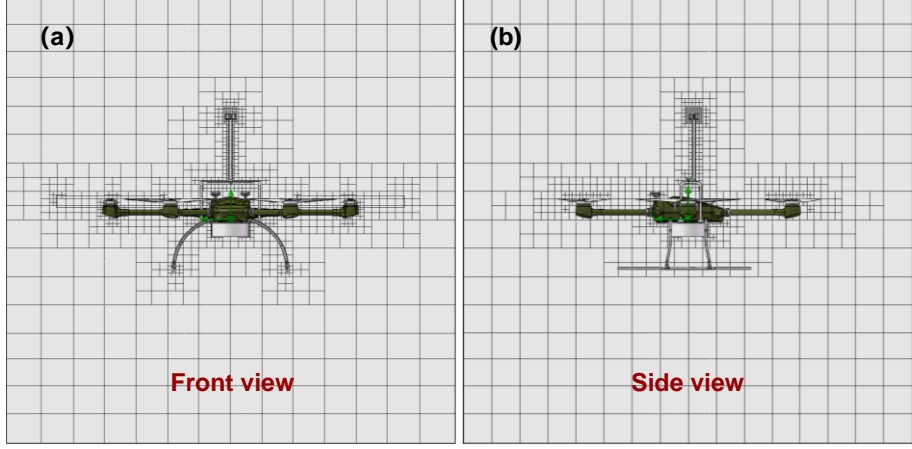

**Figure 3:Grid configuration of the computational domain.**
**Table 2:Simulation parameters configuration.**

| Parameters | Content |
| --- | --- |
| Computational domain size | $3.3 \times 3.3 \times 3.3$ m³ |
| Global domain grid size | $0.23 \times 0.23 \times 0.23$ m³ |
| Subdomain grid size | $0.0125 \times 0.0125 \times 0.0125$ m³ |
| Total number of computational domain grids | $1.11 \times 10^8$ |
| Turbulence intensity | 0.1% |
| Turbulence length scale | 0.012 m |
| Roughness | 0 |
| 30 m atmospheric pressure | $1.01 \times 10^5$ Pa |
| 1000 m atmospheric pressure | $9.00 \times 10^4$ Pa |

| | |
|---|---|
| 30 m atmospheric temperature | 25 °C |
| 1000 m atmospheric temperature | 25 °C |

To ensure relatively accurate simulations, two categories of flow field properties were specified as
computational objectives prior to the start of the simulations, and the simulations were terminated upon
convergence of the simulation results for all objectives. The first category comprised global domain
computational objectives, including average total pressure ($P_G$), average velocity ($V_G$), average vertical
velocity ($V_{Gy}$), and average forward velocity ($V_{Gz}$), where the subscript $G$ denotes the global domain.
The second category consisted of subdomain computational objectives, which included the average
velocity ($V_S$), three-dimensional average speed components $V_{Sx}$, $V_{Sy}$ and $V_{Sz}$ at the anemometer position
in the simulation coordinate system. It is noteworthy that the aforementioned average values refer to the
spatial averages over the global domain or subdomain.
Upon simulation completion, these velocity components ($V_{Sx}$, $V_{Sy}$, $V_{Sz}$) were further converted to
velocity components at the anemometer sensor position ($u_{x\_sensor}$, $u_{y\_sensor}$, $u_{z\_sensor}$) in the airframe
coordinate according to the coordinate system shown in Fig. 1((a) and (b)) and Eqs. (6) - (8) below. The
converted velocities, $u_{x\_sensor}$, $u_{z\_sensor}$, $u_{z\_sensor}$, were subtracted from the airspeed (denoted as $u_{x\_air}$,
$u_{y\_air}$, and $u_{z\_air}$) setting for each CFD simulation, to estimate the false wind signals arising from the
induced flow by the UAV rotors, expressed with $\Delta u_x$, $\Delta u_y$ and $\Delta u_z$, respectively, using Eqs. (9) - (11).
$$u_{x\_sensor} = -Vs_z, \tag{6}$$
$$u_{y\_sensor} = Vs_x, \tag{7}$$
$$u_{z\_sensor} = -Vs_y, \tag{8}$$
$$\Delta u_x = u_{x\_sensor} - u_{x\_air}, \tag{9}$$
$$\Delta u_y = u_{y\_sensor} - u_{y\_air}, \tag{10}$$
$$\Delta u_z = u_{z\_sensor} - u_{z\_air}, \tag{11}$$
In other words, the false wind signals $\Delta u_x$, $\Delta u_y$ and $\Delta u_z$ are the terms that must be determined and
corrected for in the wind measurements from the UAV.
**3 Result and Discussion**
**3.1 Example Analysis of Simulation Results**
According to the Sect. 2.2, this study develops a series of simulation scenarios for the UAV digital
model under various combinations of altitude (30 and 1000 m), wind direction (tailwind, headwind, and
crosswind), ground speed (8 to 18 m/s), and wind speed (1.5 to 14 m/s). To demonstrate the flow field
characteristics around the UAV under various scenarios, one UAV hovering scenario and six
representative simulation scenarios were specifically selected for analysis as examples.
Fig. 4 presents the cross-sectional view of the velocity flow field around the UAV in a hovering
state under wind-free conditions. In this scenario, the surrounding velocity field is solely generated by
the rotational induced flow from the UAV's own rotors. The simulated 2-4 m/s airflow around the
anemometer originates exclusively from rotor rotation, demonstrating that the rotor-induced flow during
hovering inherently interferes with wind speed measurements by the anemometer.

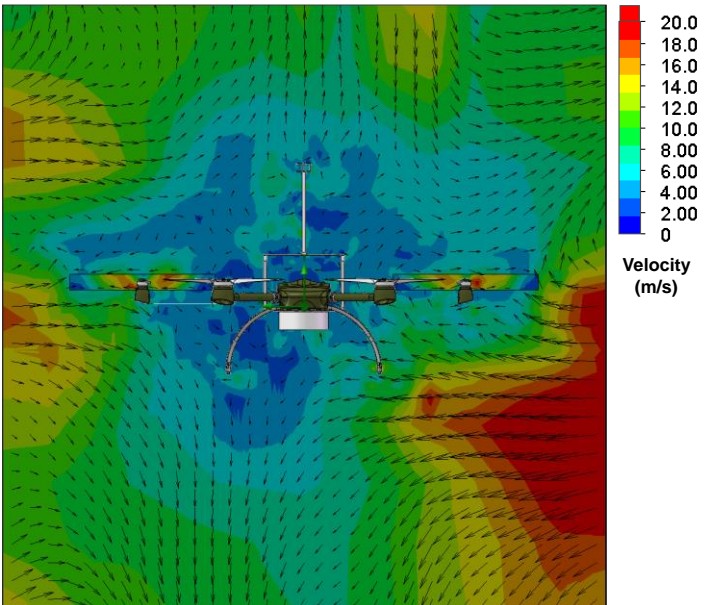


**Figure 4: The velocity flow field distribution of the UAV's hovering state.**
The other six scenarios include UAV flight simulations at altitudes of 30 m and 1000 m, with a
ground speed of 8 m/s and a wind speed of 5.4 m/s, under tailwind, headwind, and crosswind conditions.
Fig. 5 to 7 present cross-sectional views of the surrounding flow fields during UAV flight under these
conditions. In the figures, color gradients represent the magnitude of the velocity, while arrows indicate
both the direction and magnitude of the velocity. Overall, under varying wind conditions, the direction
and speed of airflow around the UAV show significant differences. While the airflow direction around
the UAV remains relatively consistent at the both simulation altitudes, the airflow speed at 1000 m is
slightly higher than at 30 m, particularly under tailwind conditions. Specifically, based on the ground
speed, wind direction, and wind speed settings, the UAV's airspeed relative to the wind is 2.6 m/s, 13.4
m/s, and 5.4 m/s in tailwind, headwind, and crosswind scenarios, respectively.

290       As show in Fig. 5 (a) and (b), in the tailwind scenario, the maximum downwash velocity occurs

directly beneath the UAV rotors, with the airflow directed vertically downward. The next highest
velocities are observed around the sides and above the rotors, where the airflow follows an inward and
downward trend. The wind speed at the anemometer location is minimally influenced by the UAV rotors,
meaning the measured wind speed represents the true airspeed. In the headwind scenario (Fig. 6 (a) and
(b)), the highest airflow velocity is detected near the area directly above the rotors, with the airflow also
following an inward and downward pattern. The lowest velocity is found directly below the rotors, where
the airflow moves upward and outward. At the anemometer's location, some interference from the UAV
rotors is present, so the wind speed at this point is a combination of the true airspeed and the rotor-
induced velocity. As exhibited in Fig. 7 (a) and (b), in the crosswind scenario with wind blowing from
left to right, the airflow around the UAV resembles that in the headwind scenario, but the overall flow
field is deflected to the right due to the crosswind, with relatively lower airflow velocity. In the scenario
with wind blowing from right to left, the flow field shifts to the left.

303       These simulation results show that the flow field around the UAV varies significantly depending

on both the presence/absence of wind and its directional characteristics, and the anemometer experiences
different levels of interference accordingly. Thus, accurately quantifying the interference of the UAV
rotors on the anemometer is essential. However, in practical application scenarios, it is also necessary to
comprehensively consider additional airflow disturbances induced by the UAV's own motion and
attitude fluctuations, and to develop corresponding dynamic compensation algorithms.

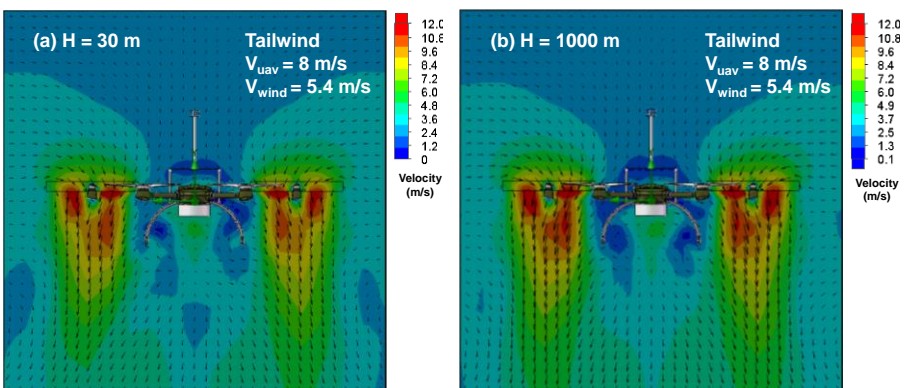

**Figure 5: Simulation flow field example results of the UAV wind measurement platform in the tailwind**
**scenario. (a) and (b) represent the longitudinal cross-sections of the simulation flow fields for the UAV at**
**altitudes of 30 m and 1000 m, with a ground speed of 8 m/s and a wind speed of 5.4 m/s.**

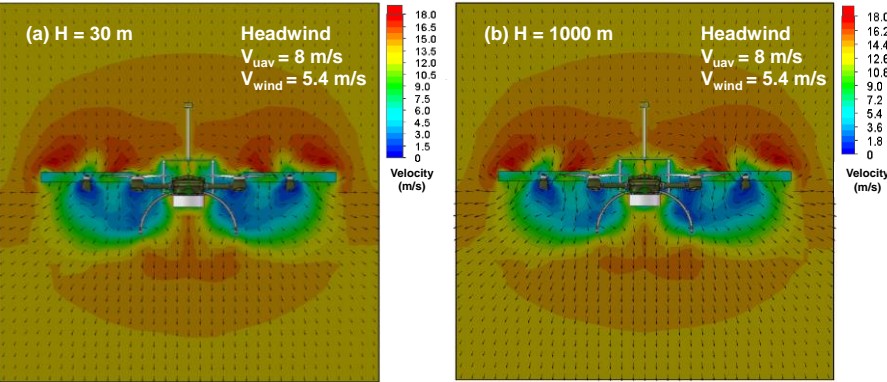

**Figure 6:Simulation flow field example results of the UAV wind measurement platform in the headwind scenario. (a) and (b) represent the longitudinal cross-sections of the simulation flow fields for the UAV at altitudes of 30 m and 1000 m, with a ground speed of 8 m/s and a wind speed of 5.4 m/s.**

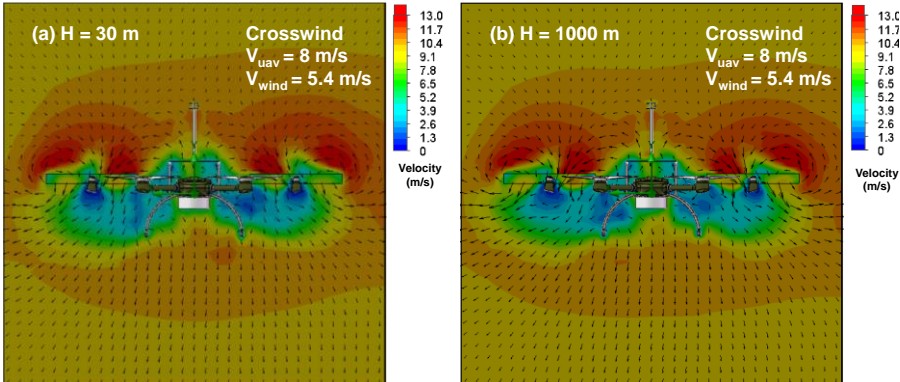

**Figure 7:Simulation flow field example results of the UAV wind measurement platform in the crosswind scenario. (a) and (b) represent the longitudinal cross-sections of the simulation flow fields for the UAV at altitudes of 30 m and 1000 m, with a ground speed of 8 m/s and a wind speed of 5.4 m/s.**

**3.2 The effect of flight altitude on rotor interference with anemometer measurements**

Through simulating the flight of UAV in all simulation scenarios, the false signals produced by the UAV rotors on the anemometer at different altitudes and wind characteristics were captured. Initially, the influence of flight altitude on the false signals was examined.

The simulated flight data under tailwind and headwind conditions were integrated into a unified data set since the UAV flight velocity vector is parallel to the tailwind and headwind velocity vectors during normal flight. The simulated false wind signals on the anemometer in the airframe $x$, $y$, and $z$ directions, caused by the propeller induced airflow under tailwind and headwind conditions, were represented by $\Delta u_x^{\mathrm{T/HW}}$, $\Delta u_y^{\mathrm{T/HW}}$, and $\Delta u_z^{\mathrm{T/HW}}$, respectively. For the tailwind and headwind datasets, according to the Wilcoxon non-parametric test for paired samples, as shown in Table 3, the differences in $\Delta u_x^{\mathrm{T/HW}}$, $\Delta u_y^{\mathrm{T/HW}}$ and $\Delta u_z^{\mathrm{T/HW}}$ were not significant ($p < 0.05$) at either the 30 m or the 1000 m

altitudes. Therefore, in the presence of tailwind or headwind, the interference from the UAV propeller-induced flow on the anemometer measurement can be considered independent of the flight altitude in this altitude range.

Similarly, the simulated false wind signals for the crosswind conditions on the anemometer in the $x$, $y$, and $z$ directions were represented by $\Delta u_x^{CW}$, $\Delta u_y^{CW}$, and $\Delta u_z^{CW}$. The Wilcoxon non-parametric test of paired samples was also applied (shown in Table 1) between the two altitudes. No significant differences were found for $\Delta u_x^{CW}$, $\Delta u_z^{CW}$ between the two altitudes, but there was an obvious discrepancy for $\Delta u_y^{CW}$($p = 0.00$)at the two altitudes. This indicates that under cross wind conditions, the disturbances of the UAV propeller in the $x$ and $z$ directions of the anemometer are not altitude dependent, but that in the $y$ direction it is necessary to distinguish the altitude. This behavior can be attributed to differences in the interaction between the y direction component and rotor rotational momentum caused by variations in atmospheric density at different altitudes under crosswind conditions.

**Table 3: Wilcoxon nonparametric tests for paired samples of false wind velocity signals between 30 m and 1000 m flight altitudes.**

| Wind Types | False Wind Signal | Significance | α | Test results |
| --- | --- | --- | --- | --- |
| Tailwind/Headwind | $\Delta u_x^{T/HW}$ | 0.93 | 0.05 | No difference |
| | $\Delta u_y^{T/HW}$ | 0.72 | 0.05 | No difference |
| | $\Delta u_z^{T/HW}$ | 0.21 | 0.05 | No difference |
| Crosswind | $\Delta u_x^{CW}$ | 0.36 | 0.05 | No difference |
| | $\Delta u_y^{CW}$ | 0.00 | 0.05 | Significant difference |
| | $\Delta u_z^{CW}$ | 0.81 | 0.05 | No difference |

**3.3 Rotor Interference on Anemometer Measurements**

This study employs a regression fitting to explore the relationship between the false wind signals generated by the UAV rotors airflow and the UAV's airspeed. Under tailwind and headwind conditions, the false wind signals ($\Delta u_x^{T/HW}$, $\Delta u_y^{T/HW}$, and $\Delta u_z^{T/HW}$) on the anemometer resulting from the UAV rotor -induced flows at both flight altitudes were aggregated and fitted as dependent variables in a regression using $u_{x\_sensor}$ as the independent variable. As shown in Fig. 8 (a), (b) and (c), good linear relationships were found between $\Delta u_x^{T/HW}$, $\Delta u_y^{T/HW}$, and $\Delta u_z^{T/HW}$ and the simulated velocity

components in the $x$-direction ($u_{x\_sensor}$), respectively. The specific relationship is described by Eqs. (12) to (14). Thus, using the UAV velocity components in $x$ direction, the false wind signals caused by the UAV propellers can be determined and removed from the raw measured wind velocity from the anemometer.

For crosswind conditions, regressions were fitted with false wind signals ($\Delta u_x^{CW}$ and $\Delta u_z^{CW}$) as dependent variables and $u_{x\_sensor}$ as the independent variable in the same way (See Fig. 9). A linear relationship was observed between the false wind signals in both $x$ and $z$ directions ($\Delta u_x^{CW}$ and $\Delta u_z^{CW}$) and $u_{x\_sensor}$, with the specific expressions in Eq. (15) and (16), respectively. As described in Sect. 3.2, $\Delta u_y^{CW}$ was sensitive to flight altitude under crosswind conditions, hence $\Delta u_y^{CW}$ at 30 m and 1000 m altitude ($\Delta u_{y(30)}^{CW}$ and $\Delta u_{y(1000)}^{CW}$) were regressed against $u_{y\_sensor}$ for the two flight altitudes separately. The $\Delta u_{y(30)}^{CW}$ exhibited a linear relationship with $u_{y\_sensor}$, as shown in Eq. (17). However, the correlation coefficient between $\Delta u_{y(1000)}^{CW}$ and $u_{y\_sensor}$ was found to be lower than 0.5, indicating that $\Delta u_{y(1000)}^{CW}$ may be considered independent of $u_{y\_sensor}$. Therefore, the average value of $\Delta u_{y(1000)}^{CW}$ (0.006 m s$^{-1}$) was regarded as the $\Delta u_{y(1000)}^{CW}$ at this flight altitude.

Despite the dependence of $\Delta u_y^{CW}$ on flight altitudes, $\Delta u_{y(30)}^{CW}$ and $\Delta u_{y(1000)}^{CW}$ are confined to a similar numeric range. Therefore, they may be roughly considered as representing $\Delta u_y$ for lower altitude (e.g., 0 to 500 m) and higher altitude (e.g., 500 to 1000 m), respectively.

Hence, for crosswind situations, the wind velocities in the $x$, $y$ and $z$ directions measured by the anemometer are corrected by subtracting $\Delta u_x^{CW}$, $\Delta u_z^{CW}$ and $\Delta u_{y(0-500)}^{CW}$ which are estimated from $u_{x\_sensor}/u_{y\_sensor}$, or at relatively high flight altitudes using a constant value of 0.006 m s$^{-1}$ for $\Delta u_{y(501-1000)}^{CW}$.

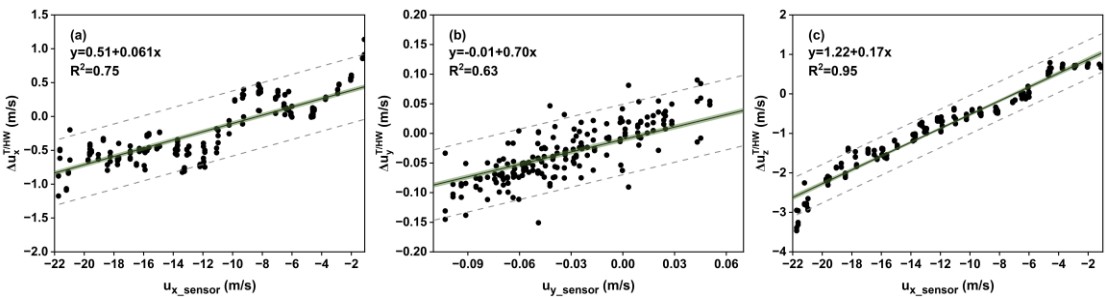

**Figure 8: Regression fit of artificial velocity ($\Delta u_x^{T/HW}$, $\Delta u_y^{T/HW}$ and $\Delta u_z^{T/HW}$) with $u_{x\_sensor}$ for tailwind and headwind flight conditions at two altitudes. In the figure, simulation data are marked with black dots, fitted curves are indicated in black lines, the 95% confidence bands are identified as green shadows, and the 95% prediction bands are represented with gray dashed area.**

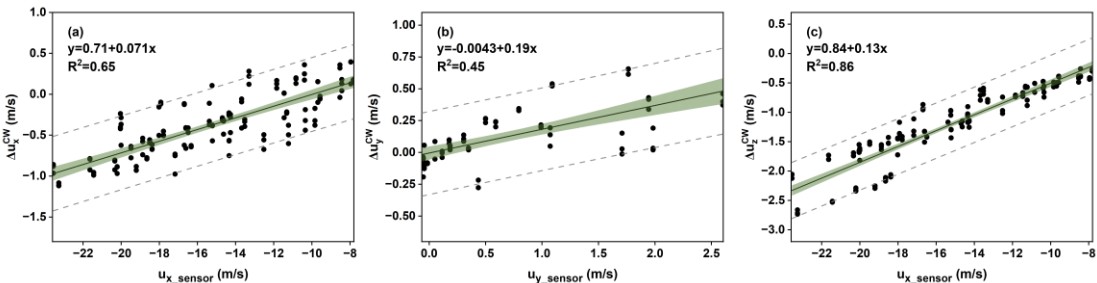


**Figure 9: Regression fit of false wind velocity signals $\Delta u_x^{CW}$, $\Delta u_z^{CW}$ and $\Delta u_{y(0-500)}^{CW}$ with $u_{x\_sensor}/u_{y\_sensor}$**
**for crosswind flight conditions at two altitudes. The symbols in the figure are the same as in Figure 6.**
$$\Delta u_x^{T/HW} = 0.51 + 0.061 \times u_x, \tag{12}$$
$$\Delta u_y^{T/HW} = -0.01 + 0.70 \times u_y, \tag{13}$$
$$\Delta u_z^{T/HW} = 1.22 + 0.17 \times u_x, \tag{14}$$
$$\Delta u_x^{CW} = 0.71 + 0.071 \times u_x, \tag{15}$$
$$\Delta u_z^{CW} = 0.84 + 0.13 \times u_x, \tag{16}$$
$$\Delta u_{y(0-500)}^{CW} = -0.0043 + 0.19 \times u_y, \quad (h = 0 \sim 500 \text{ m}), \tag{17}$$
$$\Delta u_{y(501-1000)}^{CW} = 0.006, \quad (h = 501 \sim 1000 \text{ m}), \tag{18}$$
In Eq. (17) and (18), the variable $h$ represents the flight altitude of the UAV.
**3.4 The Overall Correction Algorithm**
**3.4.1 Motion and Attitude Compensation Correction of UAV**
In addition to the false wind signals caused by propeller rotations, additional false wind velocity
signals from the anemometer can be attributed to UAV movement and attitude (pitch, roll and yaw)
changes during flight, and as such also need correction. When the UAV moves horizontally and vertically
relative to the ground, the velocity vector measured by the anemometer is a vector combination of the
true wind velocity and the UAV's ground velocity. Consequently, the ground velocity of the UAV ($v_x$ and
$v_z$, with $v_y$ always 0 due to no motion in the y direction) contributes false wind velocity components to
measurements by the anemometer. Moreover, the UAV's flight attitude undergoes adjustments in the
pitch, roll, and yaw Euler angles ($\theta$, $\varphi$, and $\psi$, respectively), in order to compensate for aerodynamic
resistance or adapt to flight plans. These adjustments lead to the anemometer measuring additional
velocities resulting from the rotational rates of the attitude angles ($\mu(\theta)$ and $\mu(\varphi)$, with $\mu(\psi)$
remaining zero due to the alignment of the rotational axis of $\psi$ with the line connecting the UAV's center
of gravity and the anemometer. Furthermore, the effect is further amplified by the distance ($r$) between

the anemometer and the UAV's center of gravity. It is noteworthy that there is currently no reported

correction algorithm for influence of attitude angle variations on anemometer wind velocity

measurements from UAVs. To obtain accurate wind information, after eliminating the aforementioned

interferences, the wind velocities ($u_x$, $u_y$ and $u_z$) observed by the anemometer in the airframe coordinate

($x$, $y$ and $z$ directions) were transformed to the North-East-Down (NED) ground coordinate using the

direction cosine matrix (DCM) as given in Eq. (19).

$$\begin{bmatrix} u_N \\ u_E \\ u_D \end{bmatrix} = \text{DCM}(\theta, \varphi, \psi) \left( \begin{bmatrix} u_x \\ u_y \\ u_z \end{bmatrix} + \begin{bmatrix} v_x \\ 0 \\ -v_z \end{bmatrix} + \begin{bmatrix} \mu(\theta) \\ -\mu(\varphi) \\ 0 \end{bmatrix} \right), \tag{19}$$

$$\text{DCM}(\theta, \varphi, \psi) = \begin{bmatrix} \cos(\psi) & -\sin(\psi) & 0 \\ \sin(\psi) & \cos(\psi) & 0 \\ 0 & 0 & 1 \end{bmatrix} \begin{bmatrix} \cos(\theta) & 0 & \sin(\theta) \\ 0 & 1 & 0 \\ -\sin(\theta) & 0 & \cos(\theta) \end{bmatrix} \begin{bmatrix} 1 & 0 & 0 \\ 0 & \cos(\varphi) & -\sin(\varphi) \\ 0 & \sin(\varphi) & \cos(\varphi) \end{bmatrix}, \tag{20}$$

where DCM is defined by Eq. (20); $u_N$, $u_E$ and $u_D$ refer to corrected North, East and Down

components of wind velocity in the ground coordinate; $v_x$ and $v_z$ are the motion velocities of the UAV

in the x and z directions respectively, which are directly provided by the GPS receiver output of the

UAV or can be directly computed from the UAV longitude/latitude coordinate output; $\mu(\theta)$ and $\mu(\varphi)$

represent the product of the pitch rate $\omega(\theta)$ and roll rate $\omega(\varphi)$, respectively, with the rotation radius $r$,

which is the distance between the anemometer and the center of gravity of the UAV, as defined in Eqs.

(21) and (22). Due to the alignment of the anemometer's z-axis with that of the UAV, the variation in

yaw ψ does not introduce false wind speed to signals from the anemometer in the airframe coordinate,

resulting in μ(ψ) being equal to zero.

$$\mu(\theta) = \omega(\theta) \times r = \frac{d(\theta)}{dt} \times r, \tag{21}$$

$$\mu(\varphi) = \omega(\varphi) \times r = \frac{d(\varphi)}{dt} \times r, \tag{22}$$

where $\omega(\theta)$ and $\omega(\varphi)$ are defined as the differentiation of $\theta$ and $\varphi$ with respect to time $t$,

respectively.

**3.4.2 Compensation Correction for Induced-Flow Disturbance by UAV Rotors**

Based on the statistical analyses of the fluid simulation results in Sect. 3.3, the regression

relationships between the false wind velocity signals generated by the propeller rotation and the

simulated wind components sensed by the anemometer are integrated into the motion and attitude

correction algorithm of UAV given in Eq. (19). The updated wind velocity correction algorithm is given
as Eq. (23), whose second and third vectors on the right side of Eq. (23) represent the contributions of
the propeller-induced wind signals under tailwind/headwind and crosswind conditions to $u_x$, $u_y$ and $u_z$,
respectively, with A and B defined in Eqs. (24) and (25) to quantify their magnitudes. Since the measured
wind velocities $u_x$ and $u_y$ from the anemometer correspond to the simulated $u_{x\_sensor}$ and $u_{y\_sensor}$,
respectively, the regression relationships are modified by replacing $u_x$ and $u_y$ with $u_{x\_sensor}$ and $u_{y\_sensor}$,
respectively. This yields the estimations of the false wind velocity signals, $\Delta u_x$, $\Delta u_y$ and $\Delta u_z$, under
different wind directions, in relation to $u_x$ and $u_y$, as specified by Eqs. (12) - (18). Using Eq. 16, the actual
wind velocity components, including north wind ($u_N$), east wind ($u_E$), and vertical wind ($u_D$), are
computed after correcting for the effects of UAV's rotor propeller disturbance, motion, and attitude on
the wind signal measurements from the anemometer.
$$
\begin{bmatrix} u_N \\ u_E \\ u_D \end{bmatrix} = \mathrm{DCM}(\theta, \varphi, \psi) \left( \begin{bmatrix} u_x \\ u_y \\ u_z \end{bmatrix} - \begin{bmatrix} A \times \Delta u_x^{T/HW} \\ A \times \Delta u_y^{T/HW} \\ A \times \Delta u_z^{T/HW} \end{bmatrix} - \begin{bmatrix} B \times \Delta u_x^{CW} \\ B \times \Delta u_y^{CW} \\ B \times \Delta u_z^{CW} \end{bmatrix} + \begin{bmatrix} v_x \\ 0 \\ v_z \end{bmatrix} + \begin{bmatrix} -\mu(\theta) \\ \mu(\varphi) \\ 0 \end{bmatrix} \right)
$$
$$(23)$$

$$A = \left| \frac{u_x}{\sqrt{u_x{}^2 + u_y{}^2}} \right|,$$
$$(24)$$

$$B = \left| \frac{u_y}{\sqrt{u_x{}^2 + u_y{}^2}} \right|,$$
$$(25)$$

**3.5 Validation of the Correction Algorithm**
A comparative experiment was designed to verify the effectiveness of the correction algorithm
described in Eq. (23). The experiment primarily compares three different wind data: the first is the three-
dimensional wind vector corrected only for UAV motion and attitude compensation (Eq. (19) and
denoted as $\mathbf{V_O}$), the second includes additional corrections for UAV rotor interference, along with motion
and attitude compensation (Eq. (23) and denoted as $\mathbf{V_R}$), and the third is the three-dimensional wind
directly measured by the meteorological tower (denoted as $\mathbf{V_T}$). The comparison experiment was
conducted with the UAV flying wind-boxes around the 80-meter meteorological tower within the
Experimental Base of the Beijing Key Laboratory of Cloud, Precipitation and Atmospheric Water
Resources. The meteorological tower was equipped with three-dimensional ultrasonic anemometers
positioned at heights of 30, 50, and 70 m, with one anemometer in the north and one in the south (see
Fig. 10). Experiments were conducted during the daytime on July 19, 2022, with neutral atmospheric
stability to minimize thermal boundary layer effects on vertical wind variability.
The UAV flew around the tower in a box flight path at a horizontal distance of about 10 m away
from the tower, at all three heights. During these flights, the UAV maintained a commanded horizontal
speed of approximately 5 m/s, a value selected as a compromise between achieving sufficient spatial
sampling resolution and maintaining stable flight attitude control. A total of 30 independent wind-box
flights were conducted, with each altitude (30, 50 m and 70 m) sampled 10 times. Each flight lasted
approximately 13 minutes, generating over 800 valid data points per altitude. Given the potential
interference from near-surface vegetation on the 30-meter anemometer on the tower, wind velocities
acquired by the UAV at 50 and 70 m heights during steady flight intervals were analyzed herein. Using
a $3\sigma$ threshold of the mean value of the entire dataset to exclude data outliers caused by sudden gusts
or UAV maneuvers (such as turning), retaining data during steady UAV flight periods.

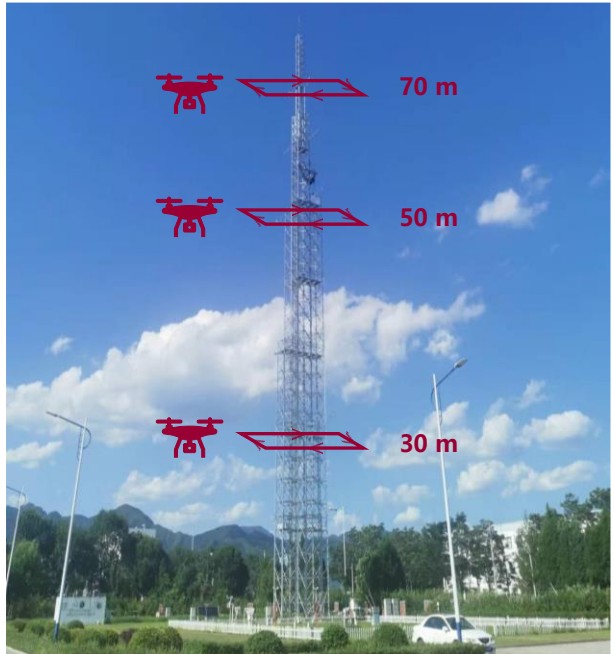

**Figure 10: Comparative experiment on wind measurements between the UAV and the meteorological tower.**
Fig. 11 presents the $V_O$, $V_R$, and $V_T$ time-series data acquired during the dual-altitude flight tests
of the UAV at 70 m and 50 m, with the 70 m altitude test data collected prior to 15:05 and the 50 m
altitude test data obtained after 15:10. The results in Fig. 11 (a) demonstrate that at elevated wind speeds
($> 3$ m s$^{-1}$), the wind velocities of $V_R$ were substantially lower than that of $V_O$. The root mean square
relative errors between $V_R$ and $V_T$, and $V_O$ and $V_T$, are 0.28 and 0.37, respectively, with the former being
approximately 24% smaller than the latter. This indicates that the correction effect of Eq. (23) is
especially pronounced in strong wind conditions. In contrast, under gentle wind speeds ($\leqslant$ 3 m s$^{-1}$), $\mathbf{V_R}$
exhibited greater consistency with $\mathbf{V_O}$ but there was still a significant down-revision in the average speed
in $\mathbf{V_R}$. The average wind speeds of $\mathbf{V_O}$, $\mathbf{V_R}$, and $\mathbf{V_T}$ were 2.4, 1.91, and 1.81 m s$^{-1}$, respectively, with $\mathbf{V_R}$
exhibiting a 22% decrease compared to $\mathbf{V_O}$. The statistical analysis using the Wilcoxon signed-rank test
confirmed a significant difference ($p < 0.01$) in wind speed between $\mathbf{V_O}$ and $\mathbf{V_T}$, whereas no significant
differences ($p > 0.01$) were found between $\mathbf{V_R}$ and $\mathbf{V_T}$. This suggests that after compensating for UAV
motion, attitude, and rotor interference, the wind speed measured by the UAV anemometer is more
closely aligned with that measured directly by the meteorological tower. Moreover, under stronger winds,
the wind direction values of $\mathbf{V_R}$, $\mathbf{V_O}$, and $\mathbf{V_T}$ were relatively similar, yet at weaker winds, $\mathbf{V_R}$ showed a
small low-bias of about 3.3% (Fig. 11 (b)). The mediocre performance of $\mathbf{V_R}$ under low wind speeds
may originate from the disruption of stable maneuverability in drone rotors caused by low wind speeds,
which in turn leads to the failure of the correction algorithm based on CFD steady-state simulations.
Fig. 12 presents the wind rose diagrams, offering a detailed overview of the wind speed and
direction characteristics for $\mathbf{V_R}$, $\mathbf{V_O}$, and $\mathbf{V_T}$. Compared to the prevailing wind direction frequency (north
wind, 39%) of $\mathbf{V_T}$, the dominant wind direction frequency errors for $\mathbf{V_O}$ and $\mathbf{V_R}$ are 40% and 5%
respectively, demonstrating the superiority of $\mathbf{V_R}$ in correcting the prevailing wind direction frequency.
Meanwhile, deviations in the secondary components introduced by $\mathbf{V_R}$ (e.g., northwester wind) indicate
directions for subsequent model optimization. These analyses indicated that Eq. (23) can effectively
correct wind measurement biases induced by UAV disturbances, motion, and attitude changes,
particularly at higher wind speeds.
In addition, it should be emphasized that while this study primarily relied on meteorological tower
data for algorithm validation, cross-validation through industrial emission scenarios has further
confirmed the algorithm's robustness in complex flow fields (Han et al., 2023).

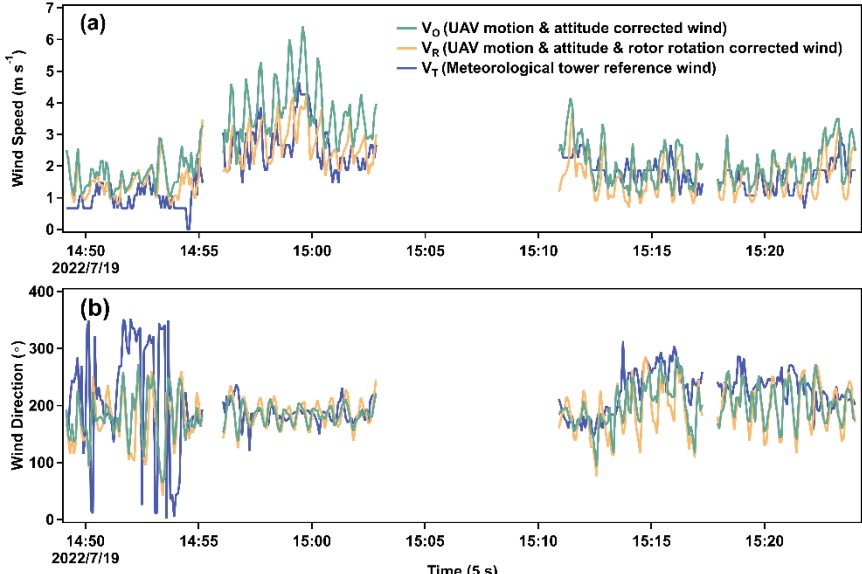


**Figure 11: Comparison of wind speed and wind direction time series for $V_R$, $V_O$, and $V_T$. (a) Comparison of wind speed time series for $V_R$, $V_O$, and $V_T$. (b) Comparison of wind direction time series for $V_R$, $V_O$, and $V_T$. (Note: The meteorological tower measured wind data at 5 s intervals, while the UAV-based measured and corrected wind data were processed with a 10 s sliding average to suppress rotor-induced high-frequency noise, followed by 5 s non-overlapping averaging to align temporally with the tower's 5 s output interval.)**

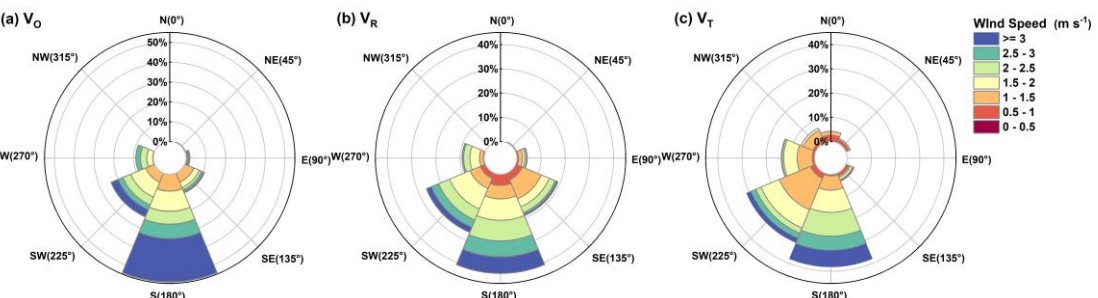

504

**Figure 12: Comparison of wind roses for $V_O$, $V_R$, and $V_T$.**

**3.6 Discussion on the Limitations of the Algorithm**

The current development of algorithms based on idealized steady-state CFD simulations relies on two key assumptions: low environmental turbulence intensity (0.1%) and turbulence length scales dominated by anemometer geometric parameters (0.012 meters). While this idealized setup effectively isolates rotor-induced flow distortion, its turbulence characteristics fundamentally differ from natural atmospheric conditions. However, it is crucial to emphasize that the algorithm's applicability under turbulent conditions remains valid. This is because rotor-induced wind speed deviations exhibit systemic long-time-scale characteristics, whereas atmospheric turbulence primarily affects measurement accuracy through random fluctuations in wind speed and direction with instantaneous nature. This

temporal-scale distinction enables our correction algorithm to effectively eliminate systemic biases while minimizing the impact of transient turbulence effects. Nevertheless, it should be noted that under stable atmospheric conditions (low wind speeds) as discussed in Sec. 3.5 or extreme weather scenarios, such airflow environments may disrupt the stable manoeuvrability of UAV rotors or obscure the systemic drainage effects of rotors, potentially leading to a nonlinear degradation in algorithm accuracy.

In addition, another limitation of our study is the assumption of a smooth surface in CFD simulations, which does not fully capture the impact of surface roughness on wind speed variations near the ground. In reality, surface roughness elements (e.g., vegetation, buildings, or terrain irregularities) alter the wind profile, increasing turbulence and wind shear in the atmospheric surface layer. This effect is particularly relevant for UAV-based wind measurements at low altitudes.

To further enhance the correction algorithm's applicability under diverse environmental conditions, future research will focus on the following aspects: conducting sensitivity studies under different turbulence intensity conditions, implementing supplementary correction modules specifically targeting atmospheric turbulence, and incorporating surface roughness length parameters in future CFD simulations. Although atmospheric turbulence presents significant challenges for UAV-based wind measurements, the correction framework established in this study has demonstrated its effectiveness in improving measurement accuracy across diverse meteorological conditions, thereby laying a critical foundation for developing reliable UAV-based wind measurement systems.

**4 Conclusions**

The scenarios involving direct measurements of wind fields within the atmospheric boundary layer using multirotor UAVs have become progressively commonplace, heightening the significance of accurate wind assessment. However, the rotor propellers during UAV flight introduce additional induced flows at the anemometer location, leading to false wind speed signals. For the present UAV-anemometer-payload configuration, a CFD-based method was used to simulate the process of the UAV wind measurement platform during stable flights under headwind, tailwind, and crosswind conditions. The analyses of induced airflows surrounding the anemometer led to a predictive tool for disturbance airflows. Building upon the UAV motion and attitude correction algorithm, a correction algorithm was proposed for the combined false wind signals from UAV rotor propeller disturbance, motion, and attitude

changes during UAV flights. Through comparison of the corrected wind speeds derived from measurements taken from the UAV platform and concurrent three-dimensional wind measurements from a nearby meteorological tower, the validity of the correction algorithm has been demonstrated. Although the algorithm still has certain limitations, it provides a feasible approach for the direct measurement of wind speed from multirotor UAV flights.

In conclusion, this study represents a significant advancement in three-dimensional wind speed measurement using UAV platforms, presenting a practical and effective method for direct and accurate wind measurement. This technological breakthrough not only creates a strong foundation for precise wind field measurements with UAVs but also provides potential avenues for the accurate quantification of gaseous pollutant emissions based on UAV. The outcomes of this work carry considerable scientific importance and offer valuable practical applications.

**Data availability.** Data are available from the corresponding author upon reasonable request.

**Author contributions.** YY conducted CFD simulations, performed data analysis, and drafted the initial manuscript. YZ contributed to the development of calibration algorithms and provided algorithm programming. YY and TH designed and conducted flight measurement validation experiments under the guidance of SML. DZ provided the site for the flight measurement validation experiments. CX, YL, and YH assisted in refining simulation scenarios and algorithms. JZ, HS, and KZ contributed to the improvement of data analysis methods. SML reviewed and edited the manuscript to ensure the accuracy and completeness of the research.

**Competing interest.** The contact author has declared that none of the authors has any competing interests.

**Acknowledgment.** The authors acknowledge financial support of the National Natural Science Foundation of China Creative Research Group Fund (grant no. 22221004).

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
