# Peer review of "A Correction Algorithm for Rotor-Induced Airflow and"

_Atmospheric Measurement Techniques, 2024_

## Referee Comment (RC2)

The manuscript entitled *"A Correction Algorithm for Rotor-Induced Airflow and Flight Attitude Changes during Three-Dimensional Wind Speed Measurements Made from a Rotary Unmanned Aerial Vehicle"* presents a novel algorithm designed to improve UAV-based wind measurements obtained through direct techniques using flow sensors. This topic is of high scientific relevance, as drone-based wind measurements can help address observational gaps within the planetary boundary layer. Furthermore, the manuscript aligns well with the scope of *Atmospheric Measurement Techniques*. However, I believe the manuscript requires further revisions before it is suitable for publication. Below, I have outlined my specific comments and suggestions for improvement.

Reviewer Comments:

In line 57, the manuscript states that indirect wind velocity estimates do not reflect flight conditions. Given extensive research on improving these methods, clarifying their specific drawbacks compared to direct measurements with airflow sensors would benefit readers.

In line 85, the manuscript notes that wind tunnel tests can improve the accuracy of airspeed-UAV motion relationships but are limited by high costs and errors from airflow reflections. However, it lacks references supporting evidence of these errors. Please include any relevant references.

In line 169, the manuscript mentions simulation parameters but does not specify the CFD framework beyond stating it is a built-in SolidWorks simulation. Clarifying the CFD framework and comparing its advantages and disadvantages in performance when compared to alternatives like Ansys Fluent would benefit the reader.

In line 181, the manuscript models the fluid as air with both turbulent and laminar flow, assuming a turbulence intensity of 0.1% and a length scale of 0.012 m. Given that atmospheric turbulence intensity ranges from 1% to 20% and length scales vary from sub-centimeter to kilometers, clarifying these assumptions would help the reader understand the limitations of the simulation results.

In line 236, the manuscript states that the wind speed at the anemometer location is minimally influenced by the UAV rotos. However, the results in Figure 9 show a significant change in measurements of wind speed and direction when the correction derived from simulation results is applied to field measurements.  In fact, this change is greater than the change observed when correcting aircraft motion alone. The manuscript should address this discrepancy in results.

In line 236, the manuscript asserts that the wind speed at the anemometer location is minimally affected by the UAV rotors. However, the results presented in Figure 9 show a noticeable alteration in both wind speed and direction when the correction derived from simulation results is applied to the field measurements. Notably, this change is more pronounced than the adjustment observed when only correcting for aircraft motion. The manuscript should thoroughly address this discrepancy and provide a clearer explanation for the observed differences in the results.

In the caption of Figure 9, it is mentioned that UAV measurements were first averaged using a 10-second sliding window before calculating 5-second averages. However, the rationale for applying a 10-second sliding average prior to computing the 5-second average is unclear. Given that moving averages can smooth out real wind fluctuations, further clarification on the necessity and impact of this approach would be beneficial to the reader.

In line 393, it is mentioned that a UAV was flown around a meteorological tower in a box pattern. However, the manuscript does not provide any information on the commanded flight speed during these experiments. Including this detail would be highly valuable for the reader, as the UAV's operating speed is a crucial parameter for understanding the validation results.

The validation results presented in Figure 9 show large errors in wind speed and wind direction estimates while operating in low wind conditions. A more thorough discussion of these errors would strengthen the contribution of this manuscript. Moreover, understanding the limitations of the presented algorithms would help the growing community of scientists using UAV-based algorithms for wind sensing assess the impact of this algorithm.

The validation results presented in Figure 9 reveal significant errors in wind speed and direction estimates, particularly under low wind conditions. A more comprehensive discussion of these errors would strengthen the manuscript by offering deeper insights into the algorithm's performance. For instance, exploring the correlation between $V_O$ , $V_R$, and $V_T$ could provide valuable context, especially given the critical role of accurate wind fluctuation estimates in turbulence measurements. Furthermore, a clearer examination of the algorithm's limitations would greatly benefit the growing community of scientists employing UAV-based wind sensing algorithms, helping them better evaluate its potential impact and applicability.

---

## Author Comment (AC1)

Response to reviewer:

We greatly appreciate the reviewer's recognition of the value and significance of the present study, as well as the very valuable comments on the paper. We have addressed the comments carefully as detailed below. The original comments are in black italic and our replies in black normal font, we also put the revised paragraph in blue after each reply to show the changes.

**Comments:**

In their manuscript Yang et al. present a correction algorithm for rotor-induced wind biases measured by a 3D anemometer placed on top of a rotary unmanned aerial vehicle. The authors used CFD simulations of an UAV under different wind conditions and UAV flight speeds to develop their algorithm. The decision to use CFD in this study allows for testing UAV under a large variety of wind conditions and with detail which would not be available if e.g. the study was conducted in an wind tunnel. The study highlights the benefits of measuring 3D wind using the UAV. This is especially important for studies in which UAVs are employed to measure (point-source) pollutant plumes to estimate emission strength, for these knowing the accurate wind speed and direction is crucial. This study brings a valuable contribution to having more accurate wind measurements from UAV platforms.

However, I do have a few major concerns regarding the methodology as well as thoroughness in discussing the possible drawbacks of their approach that I hope the authors would address.

1. The authors based their study on results from a large set of CFD simulations of a flow around a UAV with rotating blades. For these simulations they used the Solidworks model. What I lack is a reference to literature in which the model is described and also a short description of how exactly Solidworks works, what are the governing equations, why the model was chosen and what are the implications of those choices.

**Response:** We sincerely appreciate the reviewer's constructive comments and suggestions. In the revised manuscript, we have added a dedicated section titled "Section 2.1.2 Simulation Tool" to elaborate on the SolidWorks Flow Simulation tool selected for this study. This section now includes: 1) Relevant citations to support the choice of the tool, 2) A detailed description of its working principles and governing equations, 3) The reason for selecting this specific tool, and 4) The significance of this choice in the context of our work.

The specific modification is shown in the following blue text:

**"2.1.2 Simulation Tool**

The CFD simulations were conducted using SolidWorks Flow Simulation 2022, a pressure-based finite volume solver employing a fully coupled turbulence modeling approach. It employs an adaptive Cartesian mesh approach for three-dimensional solid meshing, with the governing equations being the Navier-Stokes equations for simulating the interaction of fluids, and the turbulence model utilizing the standard k- $\epsilon$

two-equation model (Jonuskaite, 2017).

The selection of SolidWorks Flow Simulation was driven by its seamless integration with CAD geometries, which eliminated potential errors associated with STL file conversions for our complex multi-rotor UAV design. Additionally, its wall functions for boundary layers effectively resolve gradient variations in boundary layers around rotating blades, reducing trial and error related to near-wall settings. The built-in solver convergence adopts a phased approach to multiple variant scenarios, decreasing the need for re-runs caused by insufficient convergence and thereby conserving computational costs. Its unique turbulence model automatically determines flow regimes (laminar, transitional, and turbulent), ensuring shorter turbulence model setup times while maintaining enhanced model accuracy (Azmi et al., 2017; Ramya et al., 2015).

While ANSYS Fluent offers advanced transient turbulence models (e.g., DES/LES), its computational cost for equivalent spatial resolution was typically higher than SolidWorks (Afaq and Ahmad, 2023). Given our need to simulate over 100 operational scenarios, SolidWorks' balance of engineering accuracy and computational tractability was deemed optimal for deriving empirical correction algorithm."

2. In Section 2.4 the authors describe their flow as "turbulent and laminar flow with turbulence intensity of 0.1% and turbulent length scale of 0.012 m" (in a 3.3X3.3x3.3 m3 domain) both of those parameters point to a laminar flow in the atmosphere. Can the authors provide a Reynolds number for their simulations? Then it would be easier to understand what kind of a flow they had in the simulations. If the flow is indeed laminar or low on turbulence, the authors should provide discussion on the implications for their study.

**Response:** Thank you for your valuable comments and for pointing out the need for a more detailed discussion of the flow conditions in our CFD simulations. We appreciate the opportunity to clarify our approach and address your concerns.

1) Calculation of Reynolds number and flow classification

To better characterize the flow regime in our study, we have calculated the Reynolds number (Re) using the standard formula:

$$Re = \frac{\rho UL}{\mu}$$

where  $\rho$  is the air density, U is the characteristic velocity, L is the characteristic length, and  $\mu$  is the dynamic viscosity of air. In our case, the characteristic length is L=0.012 m, taking the airspeed (U=13.4 m/s) of one of our simulated scenarios as the characteristic velocity. Assuming standard atmospheric conditions ( $\rho \approx 1.255$  kg/m3,  $\mu \approx 1.81 \times 10^{-5}$  Pa·s), the estimated Reynolds number is:

$$Re \approx \frac{1.255 \times 13.4 \times 0.012}{1.81 \times 10^{-5}} \approx 1.11 \times 10^{4}$$

This value is well within the turbulent flow regime, supporting the classification of the flow as turbulent rather than laminar.

2) Justification for low turbulence intensity in the CFD setup

The selected turbulence intensity (0.1%) aims to isolate the rotor-induced flow

dynamics from background atmospheric turbulence. Since the primary focus of this study is to characterize the systematic bias caused by the UAV rotor itself (rather than external atmospheric fluctuations), a low turbulence intensity was adopted to minimize confounding effects. The turbulence length scale was chosen based on the characteristic geometry of the miniature three-dimensional ultrasonic anemometer (frame width  $\sim 0.01$  m). This ensures that local vortices around the sensor are adequately captured.

We have explained the rationale for setting these parameters in Lines 223-228 of Section 2.4 in the revised manuscript, as shown below:

"The fluid was modeled as air with characteristics of turbulent and laminar flow. To isolate the rotor-induced flow dynamics from background atmospheric turbulence, a turbulence intensity of 0.1% and a turbulence length scale of 0.012 m were set. This low turbulence intensity minimizes confounding effects from ambient atmospheric fluctuations, while the length scale corresponds to the anemometer frame width (~0.01 m) to resolve rotor-generated eddies. These assumptions prioritize the systematic bias correction for rotor-induced airflow."

3) Potential impact of low turbulence intensity on results

We acknowledge that using a lower turbulence intensity than typical atmospheric conditions may have implications for the generalizability of our results. Therefore, we have added a new section, "Section 3.6 Discussion on the Limitations of the Algorithm", in the revised manuscript to specifically address the limitations of our algorithm. Within this section, we have also supplemented the limitations related to the parameter settings of this study, as illustrated below:

"Section 3.6 Discussion on the Limitations of the Algorithm

The current development of algorithms based on idealized steady-state CFD simulations relies on two key assumptions: low environmental turbulence intensity (0.1%) and turbulence length scales dominated by anemometer geometric parameters (0.012 meters). While this idealized setup effectively isolates rotor-induced flow distortion, its turbulence characteristics fundamentally differ from natural atmospheric conditions. However, it is crucial to emphasize that the algorithm's applicability under turbulent conditions remains valid. This is because rotor-induced wind speed deviations exhibit systemic long-time-scale characteristics, whereas atmospheric turbulence primarily affects measurement accuracy through random fluctuations in wind speed and direction with instantaneous nature. This temporal-scale distinction enables our correction algorithm to effectively eliminate systemic biases while minimizing the impact of transient turbulence effects. Nevertheless, it should be noted that under stable atmospheric conditions (low wind speeds) as discussed in Sec. 3.5 or extreme weather scenarios, such airflow environments may disrupt the stable manoeuvrability of UAV rotors or obscure the systemic drainage effects of rotors, potentially leading to a nonlinear degradation in algorithm accuracy.

In addition, another limitation of our study is the assumption of a smooth surface in CFD simulations, which does not fully capture the impact of surface roughness on wind speed variations near the ground. In reality, surface roughness elements (e.g., vegetation, buildings, or terrain irregularities) alter the wind profile, increasing turbulence and wind shear in the atmospheric surface layer. This effect is particularly relevant for UAV-based wind measurements at low altitudes.

To further enhance the correction algorithm's applicability under diverse environmental conditions, future research will focus on the following aspects: conducting sensitivity studies under different turbulence intensity conditions, implementing supplementary correction modules specifically targeting atmospheric turbulence, and incorporating surface roughness length parameters in future CFD simulations. Although atmospheric turbulence presents significant challenges for UAVbased wind measurements, the correction framework established in this study has demonstrated its effectiveness in improving measurement accuracy across diverse meteorological conditions, thereby laying a critical foundation for developing reliable UAV-based wind measurement systems."

3. In their results authors present time averaged wind fields from which they derive their correction algorithm. However, the actual measured wind is highly fluctuating and dependent on the atmospheric stability and, if close to the surface, the surrounding orography and obstacles. How do stationary results relate to turbulent atmospheric measurements? Additional discussion on influence of turbulence on UAV wind measurements would be beneficiary.

**Response:** We appreciate the reviewer for in-depth reflection and valuable suggestions on our work. We acknowledge that atmospheric wind fields in practical measurements often exhibit turbulent characteristics, where wind speed may demonstrate pronounced pulsating features due to atmospheric stability, terrain, and obstacles.

Our correction algorithm is constructed based on steady flow field analysis derived from CFD simulations. This methodology allows us to isolate and quantify systematic errors induced by rotor effects without being directly affected by random turbulence during measurements. In wind speed measurement applications, correcting such systematic errors is critical as it compensates for inherent disturbances from the UAV platform, thereby aligning measurement results more closely with true atmospheric wind speeds.

While our simulations assume steady-state conditions, our calibration method remains applicable in turbulent environments. This is because, even under turbulence, the fundamental structure and magnitude of rotor-induced flows are predominantly governed by the UAV's flight attitude and dynamic parameters rather than instantaneous turbulent disturbances. Turbulence-induced wind speed variations typically manifest as random short-term fluctuations, whereas our calibration focuses on systematic biases caused by the UAV's aerodynamic effects, which remain stable over longer timescales. Consequently, the application of this algorithm still effectively enhances wind speed measurement accuracy under turbulent conditions.

Furthermore, in turbulent environments, uncertainties in UAV-based wind speed measurements arise not only from rotor-induced flows but also from factors such as sensor response time, data sampling frequency, and UAV attitude variations. To further enhance measurement reliability in turbulent environments, future research will focus on the following aspects: conducting sensitivity studies under different turbulence intensity conditions, developing dynamic CFD models for unsteady flow fields, and implementing supplementary correction modules specifically targeting atmospheric turbulence.

We appreciate the reviewer's suggestions and have supplemented the discussion on turbulence effects on UAV wind speed measurements in the revised manuscript (Section 3.6, Lines 506-533) to further refine the applicability analysis of this study, as shown in the blue text in the reply to comment 2.

4. I believe the simulations in this study had neutral stratification (it is not explicitly stated). However, these conditions are rarely encountered in the real atmosphere. How applicable is this algorithm for when, for example, the atmosphere is unstable and the vertical wind component is much stronger? Can authors provide a discussion on potential errors which may arise when applying this algorithm.

**Response:** We sincerely appreciate the reviewer's insightful comment regarding the applicability of our correction algorithm under non-neutral atmospheric stratification conditions. We fully agree that real-world atmospheric turbulence (particularly in unstable conditions with strong vertical winds) poses additional challenges for UAV-based wind measurements.

As stated in the response to previous Comment 3, this study employs CFD numerical simulations based on idealized steady-state conditions, primarily focusing on investigating the inherent drainage effects of UAV rotors. It should be noted that the current numerical experimental design does not fully account for the prevalent non-stationary turbulent structures and significant vertical wind component conditions in actual atmospheric environments. This methodological choice is primarily based on the following considerations: Firstly, steady-state conditions can effectively isolate environmental turbulence interference; Secondly, the time scale of rotor-induced drainage fundamentally differs from that of atmospheric turbulence, and this distinction ensures the algorithm's applicability under conventional meteorological conditions.

It should be emphasized that under extreme weather conditions (e.g., high turbulence intensity or strong vertical winds), such complex airflow environments may obscure the systematic drainage effects of rotors. Specifically, potential overlaps between turbulent energy spectra and rotor characteristic frequencies, along with momentum transport interference caused by vertical wind fields, might lead to a nonlinear degradation pattern in the algorithm's accuracy. However, the precise magnitude of accuracy reduction requires quantitative evaluation through subsequent meticulously designed experiments.

Additionally, the discussion of potential errors which may arise when applying this algorithm in the revised manuscript is also presented in the response to Comment 2 as shown in blue text.

5. The authors simulate flow at 30 m and 1000 m heights and recommend their correction algorithm for UAV flights below and above 500 m, respectively. However, wind variability is highest within the atmospheric surface layer (typically below 500 m). Why were only these two heights chosen? Would additional simulations closer to

**the surface give more realistic assessment of near-surface conditions? The influence of surface roughness and stability effects should be discussed.**

**Response:** Thank you for your insightful comment. The selection of 30 m and 1000 m heights was intended to represent two distinct atmospheric conditions: nearsurface conditions within the atmospheric boundary layer (30 m) and free-atmosphere conditions (1000 m). The lower height (30 m) represents a typical operating height for industrial UAV applications, where rotor-induced airflow distortion is most pronounced due to higher air density. The upper height (1000 m) was chosen to characterize free atmospheric flow patterns under reduced density effects.

We acknowledge that wind variability is highest within the atmospheric surface layer (typically below 500 m), and additional simulations at intermediate heights (e.g., 50 m or 100 m) could provide further insights into near-surface conditions. However, our primary objective was to develop and validate a rotor-induced airflow correction algorithm rather than to comprehensively model the entire atmospheric boundary layer. The current heights were selected to capture the key differences between low-altitude and high-altitude UAV measurements while balancing computational feasibility.

Regarding the influence of surface roughness and stability effects, our CFD simulations without explicitly considering these factors. We recognize that surface roughness and atmospheric stability can impact wind speed profiles and turbulence characteristics, which in turn affect UAV-based wind measurements. Future studies could incorporate boundary-layer parameterization schemes or large-eddy simulations (LES) to better assess these influences.

We have clarified the reasons for choosing two heights in Section 2.2 (Lines 171-174) of the revised manuscript, as follows:

"For the UAV flight simulation, we considered over a hundred flight envelope scenarios, including parameters such as UAV flight altitude, wind direction, and wind speed. Since the UAV's predominant flights are within the atmospheric boundary layer, characterized by significant variability in wind speed and directions, a flight envelope for the UAV in the simulated environments was setup for the complete UAV digital model for flight altitudes of 30 and 1000 meters, respectively. The lower height (30 m) represents the typical operational altitude for industrial UAV applications within the boundary layer, while the upper height (1000 m) corresponds to the altitude where atmospheric flow transitions to more stable, low-density free-stream conditions."

Furthermore, we have added a sentence in the Section 2.4 (Lines 222-223) of the manuscript to illustrate the wall (including the surface) conditions assumed by the current simulation, as follows:

"The total number of grids in the computational domain was  $1.11 \times 10^8$ , and the specific grid configurations are shown in the Fig. 2. The wall is set as an adiabatic wall, and its roughness is set to 0."

We have already discussed the effects of surface roughness in Section 3.6 (Lines 521-525) of the revised manuscript, as shown in the blue text in the reply to comment 2.

6.In connection to the previous comment: the authors validate their results by

measuring wind using the UAV and comparing results with the measurements from a near-by meteo tower which measures winds at 3 different heights. Firstly, they exclude from their analysis the wind measurements at 30 m due to the influence of orography even though their correction algorithm is based on simulations at 30 m. I think this should be properly addressed in the text. Secondly, the results are presented in Fig. 9 which seems to be showing wind speed and direction comparison at only one height. If the measurements are somehow aggregated, it should be clearly stated how. If not, then the height at which the measurements are compared should be indicated in the text and why the second is left out should be discussed. Preferably wind at all 3 heights should be compared and shown in the results section. Lastly, the authors state that their results show especially good fit for higher wind speeds (they define them as  $\geq 3$  m/s, why this threshold?). From Fig 9. It seems that for approximately 20% of data this condition is met. And they present two sets of data of about 15 min with 5 s sampling frequency. Can the authors comment on the statistical robustness of these conclusions and possible implications.

**Response:** We sincerely appreciate the reviewer's valuable comments. Regarding the three key concerns you raised, we provide the following clarifications:

1) Clarification regarding data screening at 30 m altitude

This study excluded wind speed measurements at the 30 m altitude during data analysis, based on the following considerations: The average canopy height in the experimental area was approximately 25~28 m. When the UAV hovered at the 30 m altitude, its rotor blades maintained a mere 2~5 m clearance above the canopy top. At this proximity, canopy turbulence induces significant near-canopy turbulence effects, resulting in systematic deviations between the wind field characteristics at this altitude and those observed at the same height by the meteorological tower. To uphold the principle of homogeneity in comparative studies, after extensive deliberation by the research team, this altitude layer was ultimately excluded from the formal experimental dataset.

2) Clarification on data presentation in Figure 9 of the original manuscript

Upon review, we confirm that the original Figure 9 (Figure 11 in the revised manuscript) indeed presents comparative data for the 50-meter and 70-meter altitude layers. We acknowledge that insufficient clarity in our initial description may have caused misinterpretation. To enhance transparency, we have supplemented the following statement in Lines 467-469 of Section 3.5 in the revised manuscript:

"Fig. 11 presents the  $V_0$ ,  $V_R$ , and  $V_T$  time-series data acquired during the dualaltitude flight tests of the UAV at 70 m and 50 m, with the 70 m altitude test data collected prior to 15:05 and the 50 m altitude test data obtained after 15:10."

3) Supplementary argument on algorithm robustness

We appreciate your professional scrutiny regarding the comprehensiveness of model validation. This study indeed faced inherent constraints in data acquisition, resulting in a direct validation sample size below ideal expectations. To multidimensionally demonstrate algorithm reliability, we have supplemented the following cross-validation research.

In our companion study published in Atmospheric Measurement Techniques (DOI:

10.5194/amt-17-677-2024), the research team applied this algorithm to monitor industrial emission fluxes. Utilizing a mass balance-based flux inversion model with algorithm-corrected wind speed data and synchronized CO2 concentration measurements (via UAV-mounted CRDS analyzer), we successfully quantified CO2 emission fluxes from coke oven batteries in a steel plant. Critical validation metrics reveal: drone-derived flux estimates (110  $\pm$  18 t/h) show a high consistency with material balance calculations (103  $\pm$  32 t/h), demonstrating <15% relative deviation. Considering the sensitivity of emission flux calculation to wind velocity measurement errors, these empirical results substantiate the algorithm's effectiveness in real-world wind speed correction.

Furthermore, we have added the following discussion at the conclusion of Section 3.5 (Lines 493-495):

"In addition, it should be emphasized that while this study primarily relied on meteorological tower data for algorithm validation, cross-validation through industrial emission scenarios has further confirmed the algorithm's robustness in complex flow fields (Han et al., 2023)."

As I mentioned above I miss general discussion of limitations and drawbacks in this study. Neither in results or in the conclusions sections are these properly discussed.

**Minor comments:**

7. Line 130 (And table 1.) If the UAV can only operate under the true WS of  $\leq 18$  m/s, then doesn't the e.g. fly speed of 14 m/s with 10.7 m/s tail wind exceed that?

**Response:** We sincerely appreciate the reviewer's meticulous comments. The maximum flight speed of the UAV we adopted is 18 m/s, which refers specifically to its ground speed. In the design of our simulated flight scenarios, the maximum designed ground speed is indeed set to 18 m/s, which aligns with the operational speed range of the UAV.

8. Line 140 I think it should be clarified how both (wind) speeds were imposed in simulations. The domain seems too small to have the UAV to actually be moving through the simulations. But if I am wrong this should be clarified.

**Response:** We sincerely apologize for any confusion. Regarding the wind speed parameter issue raised by the reviewer, please allow us to provide supplementary clarification:

During the initial design phase of the program, we adopted multi-parameter combinations of wind speed, wind direction, and ground speed to construct simulation scenarios. In the actual implementation of simulations, we converted wind speed and ground speed into airspeed through vector synthesis for simulation calculations. This dual parametrization approach was primarily employed to ensure comprehensive coverage of the flight envelope in our simulation scenarios.

To enhance the clarity of presentation, we have added the following content in Section 2.2 (Lines 180-181) of the revised manuscript:

"It should be noted that the numerical simulations were conducted by converting

**wind speed and ground speed into airspeed through vector synthesis."**

We also thank the reviewer for prompting us to clarify the rationale behind the setup of the computational domain. We designed this computational domain primarily because our study focuses on the flow field around the anemometer. Within this domain, the UAV can fully and normally perform rotor rotation simulations and attitude changes.

In the revised manuscript, we have added the following explanation in blue text in Section 2.4 (Lines 213-215) to clarify the rationale for parameter selection:

"The simulation parameters primarily include the computational domain and mesh, fluid and environmental properties, as well as the rotating region. During the CFD flow simulations of the UAV using Solidworks, the computational domain dimensions (3.3  $\times$  3.3  $\times$  3.3 m3) were determined by prioritizing the analysis of flow field distribution around the anemometer while balancing computational costs."

9. Line 149-164 It would be good to show the forces acting on the UAV in a diagram. Also it would be good to have a diagram of the UAV, the projection surfaces and the angles which are mentioned in the equations (1)-(5).

**Response:** We sincerely appreciate the reviewer's constructive suggestion. In Section 2.3 (Lines 191-192) of the revised manuscript, we have added a figure to illustrate the forces acting on the UAV, as shown below:

"Figure 2: Schematic diagram of forces acting on a UAV."

10. Line 165 This paragraph is very difficult to decipher. Is "that" in the first sentence extra?

**Response:** We sincerely apologize for any confusion caused to the reviewer. We have revised the description of the sentence in question, and it now appears in the revised manuscript (Section 2.3, Lines 206-209) as shown below:

"The complete flight envelope was defined by combinations of critical parameters, including wind directions, wind speeds, airspeeds, ground speed, inclination, wind resistance, pull, and M. A series of CFD simulations were conducted to systematically evaluate the simulated wind field characteristics for each unique parameter set within this envelope."

11. Line 167 Since the authors make a clear distinction between the many different wind vectors (wind from UAV movement, wind from the simulation, measured wind at the

anemometer) they should take care to clarify each time to which wind they are referring to.

**Response:** We sincerely thank the reviewer's valuable suggestion. In the revised manuscript (Section 2.3, Line 208), we have clarified the wind type, as described below:

"The complete flight envelope was defined by combinations of critical parameters, including wind directions, wind speeds, airspeeds, ground speed, inclination, wind resistance, pull, and M. A series of CFD simulations were conducted to systematically evaluate the simulated wind field characteristics for each unique parameter set within this envelope."

**12. Line 172-180 Can the simulation specifications be shown in a table for easier following?**

**Response:** We sincerely thank the reviewer's valuable suggestion. In the revised manuscript (Section 2.4, Line 242), we have added a table summarizing the simulation parameters to aid reader comprehension, which is presented below:

| Parameters                                 | Content                                          |
|--------------------------------------------|--------------------------------------------------|
| Computational domain size                  | $3.3 \times 3.3 \times 3.3 \text{ m}^3$          |
| Global domain grid size                    | $0.23 \times 0.23 \times 0.23 \text{ m}^3$       |
| Subdomain grid size                        | $0.0125 \times 0.0125 \times 0.0125 \text{ m}^3$ |
| Total number of computational domain grids | $1.11 \times 10^{8}$                             |
| Turbulence intensity                       | 0.1%                                             |
| Turbulence length scale                    | 0.012 m                                          |
| Roughness                                  | 0                                                |
| 30 m atmospheric pressure                  | $1.01 	imes 10^5$ Pa                             |
| 1000 m atmospheric pressure                | $9.00 	imes 10^4 \mathrm{Pa}$                    |
| 30 m atmospheric temperature               | 25 °C                                            |
| 1000 m atmospheric temperature             | 25 °C                                            |

**"Table 2: Simulation parameters configuration.**

13.Line 188 What is the depth of the rotating cylinders and why was it chosen?

**Response:** We sincerely appreciate the reviewer's careful scrutiny. The depth of the rotating cylinder specified in our study is 85 mm, which is determined based on the overall height of the rotor. In general, the inner edge of this rotating cylinder is positioned adjacent to the outer edge of the UAV rotor.

14. Line 195-216 This whole section is a bit confusing to read due to the amount of different velocity definitions. What exactly are global domain and subdomain? What is the difference between Vg and Vs exactly? If they are mean velocities, what is the averaging time?

**Response:** We appreciate the reviewer's feedback regarding the clarity of some definitions in Section 2.4. We acknowledge that the distinction between terms could be better articulated. Below, we provide a detailed clarification:

1) Definition of global domain and subdomain

Global Domain refers to the entire CFD simulation space encompassing the rotary UAV and its rotor-induced airflow. It captures the large-scale flow interactions between the UAV and ambient wind.

Subdomain is a refined, high-resolution region nested within the global domain, defined by the location and size of the anemometer. Its configuration is designed to accurately simulate the impact of the additional airflow caused by the rotor on the anemometer.

In the original manuscript, Lines 174-180 contain a description of the global domain and subdomain.

2) Difference between  $V_g$  and  $V_s$ :

 $V_g$  (Global Domain Velocity) represents the ambient wind velocity derived from the global CFD simulation.

 $V_s$  (Subdomain Velocity) refers to the simulated velocity of the subdomain (at the anemometer location).

3) Averaging Time Clarification:

Both  $V_g$  and  $V_s$  are spatially averaged solutions obtained after multiple iterations during the CFD simulation process, rather than representing time-averaged values.

We have added a clarification in the revised manuscript (Section 2.4, Lines 250-251) to specify that these are spatially solved values, in order to avoid confusion among readers. The updated text reads as follows:

"It is noteworthy that the aforementioned average values refer to the spatial averages over the global domain or subdomain."

**15. Line 197 What does "convergence of the simulation results" mean exactly?**

**Response:** We thank the reviewer for raising this important point regarding the clarification of "convergence of the simulation results". Below, we provide a detailed explanation of the convergence criteria used in our CFD simulations.

In the context of our CFD simulations, "convergence" refers to the state where the iterative numerical solution of the governing equations (e.g., Navier-Stokes equations) reaches a stable equilibrium. This is achieved when: The residuals (i.e., imbalances in mass, momentum, and energy equations) decrease to a predefined threshold (e.g., below  $10^{-5}$  for scaled residuals). Key physical quantities (e.g.,  $V_g$ ,  $V_s$ ) exhibit negligible variation (e.g.,

Figure 5: The velocity flow field distribution of the UAV's hovering state."

18. Line 275 A "for the cross-wind conditions" is missing from the sentence?

**Response:** We sincerely appreciate the reviewers' thorough review. In this sentence, we indeed omitted "for the crosswind conditions". We have made the necessary revision in the corresponding location (Lines 333-334) of the revised manuscript, as shown below:

"Similarly, the simulated false wind signals for the crosswind conditions on the anemometer in the x, y, and z directions were represented by  $\Delta u_x^{CW}$ ,  $\Delta u_y^{CW}$ , and  $\Delta u_z^{CW}$ ."

**19. Line 280 Can the authors discuss why only in the crosswind there has to be height distinction for the vertical wind speed?**

**Response:** We appreciate the reviewer's insightful question. Through our examination, the parameter  $\Delta u_y^{CW}$  exhibits high sensitivity under crosswind scenarios. It is important to specifically clarify that  $\Delta u_y^{CW}$  represents the disturbance wind

component perpendicular to the UAV's heading axis within the horizontal plane. We sincerely apologize for the mislabeling of "upward" in the *y* direction within the original manuscript, which may have led readers to misinterpret it as a vertical wind disturbance component. Based on this correction, subsequent discussions will focus on elucidating why the disturbance winds in the *y* direction exhibit significant differences at varying altitudes under crosswind conditions. This phenomenon may be primarily attributed to the following reason:

In crosswind conditions, the rotor-induced airflow exhibits significant directional asymmetry in the horizontal plane. There exists an interaction between the y direction component of the crosswind (perpendicular to the flight direction axis) and the rotor's rotational momentum, which is influenced by the differing atmospheric densities between the 30 m and 1000 m altitude settings in the simulation. At 30 m altitude, the airflow demonstrates stronger interaction with UAV attitude adjustments (e.g., tilt compensation), while at 1000 m, the reduced air density mitigates these interactions. This altitude-dependent aerodynamic deformation leads to discrepancies in the simulated y direction component of the crosswind.

Additionally, we have added the following discussion in Section 3.2, Lines 339-341 of the revised manuscript to facilitate reader's understanding:

"This indicates that under cross wind conditions, the disturbances of the UAV propeller in the x and z directions of the anemometer are not altitude dependent, but that in the y direction it is necessary to distinguish the altitude. This behavior can be attributed to differences in the interaction between the y direction component and rotor rotational momentum caused by variations in atmospheric density at different altitudes under crosswind conditions."

**20. Fig 6 and 7 Why Du\_x is fitted to ux\_sensor and Du\_y is fitted to uy\_sensor but Du z is fitted to ux sensor?**

**Response:** We appreciate the reviewer's attentive observation regarding the parameter selection in Figures 9-10 (corresponding to Figures 6-7 in the original manuscript). The pairing of  $\Delta u_z$  with  $u_{x\_sensor}$  (rather than  $u_{z\_sensor}$ ) originated from our CFD simulations revealing a stronger correlation between  $\Delta u_z$  and  $u_{x\_sensor}$  compared to its correlation with  $u_{z\_sensor}$ . This phenomenon is caused by the combined effect of changes in the drone's attitude and the asymmetry in the rotor-induced airflow.

21. Line 370 Wrong reference to definition of A and B? Should be eqs. (24) and (25) instead (17) and (18)?

**Response:** We sincerely appreciate the reviewer's thorough review. There was indeed an error in the citation of the definitions of A and B in the original manuscript, and we have made the following revisions in the revised version:

"The updated wind velocity correction algorithm is given as Eq. (23), whose second and third vectors on the right side of Eq. (23) represent the contributions of the propeller-induced wind signals under tailwind/headwind and crosswind conditions to  $u_x$ ,  $u_y$  and  $u_z$ , respectively, with A and B defined in Eqs. (24) and (25) to quantify their magnitudes."

**22. Line 415 Wrong reference? Equation (23)?**

**Response:** We sincerely appreciate the reviewer's thorough review. After verification, we found that the formula citation in Line 415 is indeed inaccurate, which may have been inadvertently caused during previous revisions. In the revised version (Lines 492-494), we have corrected this citation, as follows:

"These analyses indicated that Eq. (23) can effectively correct wind measurement biases induced by UAV disturbances, motion, and attitude changes, particularly at higher wind speeds."

**23. Line 411 "small low bias" can the authors quantify this?**

**Response:** We sincerely thank the reviewer's suggestion. We have quantified the "small low bias" and made the following revisions in Section 3.5, Lines 481-482 of the revised manuscript:

"Moreover, under stronger winds, the wind direction values of  $V_R$ ,  $V_O$ , and  $V_T$  were relatively similar, yet at weaker winds,  $V_R$  showed a small low-bias of about 3.3% (Fig. 11 (b))."

24. Line 412-417 I would say all three wind speeds in Fig 10 show predominantly northernly winds. The main difference is that Vo has about 15% more wind going to the south than the wind measured at the meteo tower Vt. Vr indeed does not have those 15% bias in comparison to the Vt, however Vr has around 15% higher NW wind component that is not present in Vo and Vt. Is the Vr wind then really showing a much better agreement with the tower than Vo?

**Response:** We appreciate the reviewer's insightful observation regarding the wind direction components in Figure 12 (corresponding to Figure 10 in the original manuscript).

As shown in Figure 12, the northerly wind, being the prevailing wind direction during the flight tests (with a frequency of 40%), has observation accuracy that serves as a critical parameter for meteorological studies and atmospheric pollutant dispersion research. It is important to clarify that the core advantage of the correction algorithm proposed in this study lies in its systematic correction of the prevailing north wind direction. Specifically, the frequency error for north winds is significantly reduced from 41% in  $V_0$  to 5% in  $V_R$  after correction. Although the reviewers pointed out that the correction process may introduce biases in secondary component wind directions, such as northwester winds, the magnitude of such errors in non-prevailing wind directions and their potential impact on the overall research objectives remain far lower than the critical deviations observed in the uncorrected prevailing wind direction.

We fully understand with the reviewer's concern for non-prevailing wind directions. In fact, in Section 3.5, Lines 486-490 of the revised manuscript, we have explicitly incorporated the following addition:

"Compared to the prevailing wind direction frequency (north wind, 39%) of  $V_T$ , the dominant wind direction frequency errors for  $V_O$  and  $V_R$  are 40% and 5% respectively, demonstrating the superiority of  $V_R$  in correcting the prevailing wind

direction frequency. Meanwhile, deviations in the secondary components introduced by  $V_R$  (e.g., northwester wind) indicate directions for subsequent model optimization."

25. Fig 9 Does the meteo tower measure the instantaneous wind every 5s or it gives 5s averages? Also, why impose first the running mean on the UAV measurements before making 5s averages?

**Response:** We sincerely appreciate the reviewer's insightful questions regarding the data processing steps in Figure 11 (corresponding to Figure 9 in the original manuscript). We recognize the need for clearer documentation of temporal averaging methods. Below, we provide a detailed clarification to explicitly address these points:

First, the data measured by the anemometer on the meteorological tower is averaged every 5 seconds.

Second, we applied a 10 s sliding window prior to computing 5 s averages in Figure 11. The reasons for this are as follows. The raw wind measurements from the UAV (before correction) inherently contain high-frequency fluctuations caused by rotor-induced turbulence and rapid attitude changes. These perturbations occur at subsecond timescales, which are unrelated to atmospheric wind variability. A 10 s sliding window was selected based on spectral analysis of the raw data, as it effectively suppresses noise about 0.1 Hz while preserving the signal trends relevant to atmospheric motions. This step ensures comparability with the meteorological tower's 5 s data, which inherently lacks such high-frequency artifacts due to its stable mounting and calibrated anemometers. After noise reduction via the 10 s sliding average, we calculated non-overlapping 5 s averages to exactly match the temporal resolution of the meteorological tower measurements (5 s discrete outputs). This two-step averaging ensures both datasets share identical timestamps and statistical representativeness, enabling a fair comparison between the **Vo**, **V**R and **V**T.

Furthermore, we have made the following modifications to the title of Figure 11 in the revised manuscript:

"Figure 11. Comparison of wind speed and wind direction time series for  $V_R$ ,  $V_O$ , and  $V_T$ . (a) Comparison of wind speed time series for  $V_R$ ,  $V_O$ , and  $V_T$ . (b) Comparison of wind direction time series for  $V_R$ ,  $V_O$ , and  $V_T$ . (Note: The meteorological tower measured wind data at 5-second intervals, while the UAV-based measured and corrected wind data were processed with a 10 s sliding average to suppress rotor-induced high-frequency noise, followed by 5 s non-overlapping averaging to align temporally with the tower's 5 s output interval.)"

26. Fig 10 The caption does not match the labels (a), b), c)) in the figure.

**Response:** We sincerely appreciate the reviewer's meticulous review. In the revised manuscript, we have modified the caption of Figure 12 (corresponding to Figure 10 in the original manuscript) to align with the order of the labels in the figure, as shown below:

"Figure 12: Comparison of wind roses for Vo, VR, and VT."

---

## Author Comment (AC2)

Response to reviewer:

We greatly appreciate the reviewer's recognition of the value and significance of the present study, as well as the very valuable comments on the paper. We have addressed the comments carefully as detailed below. The original comments are in black italic and our replies in black normal font, we also put the revised paragraph in blue after each reply to show the changes.

*Comments:*

*In their manuscript Yang et al. present a correction algorithm for rotor-induced wind biases measured by a 3D anemometer placed on top of a rotary unmanned aerial vehicle. The authors used CFD simulations of an UAV under different wind conditions and UAV flight speeds to develop their algorithm. The decision to use CFD in this study allows for testing UAV under a large variety of wind conditions and with detail which would not be available if e.g. the study was conducted in an wind tunnel. The study highlights the benefits of measuring 3D wind using the UAV. This is especially important for studies in which UAVs are employed to measure (point-source) pollutant plumes to estimate emission strength, for these knowing the accurate wind speed and direction is crucial. This study brings a valuable contribution to having more accurate wind measurements from UAV platforms.*

*However, I do have a few major concerns regarding the methodology as well as thoroughness in discussing the possible drawbacks of their approach that I hope the authors would address.*

*1. The authors based their study on results from a large set of CFD simulations of a flow around a UAV with rotating blades. For these simulations they used the Solidworks model. What I lack is a reference to literature in which the model is described and also a short description of how exactly Solidworks works, what are the governing equations, why the model was chosen and what are the implications of those choices.*

**Response:** We sincerely appreciate the reviewer's constructive comments and suggestions. In the revised manuscript, we have added a dedicated section titled "Section 2.1.2 Simulation Tool" to elaborate on the SolidWorks Flow Simulation tool selected for this study. This section now includes: 1) Relevant citations to support the choice of the tool, 2) A detailed description of its working principles and governing equations, 3) The reason for selecting this specific tool, and 4) The significance of this choice in the context of our work.

The specific modification is shown in the following blue text:

"2.1.2 Simulation Tool

The CFD simulations were conducted using SolidWorks Flow Simulation 2022, a pressure-based finite volume solver employing a fully coupled turbulence modeling approach. It employs an adaptive Cartesian mesh approach for three-dimensional solid meshing, with the governing equations being the Navier-Stokes equations for simulating the interaction of fluids, and the turbulence model utilizing the standard k-ε

two-equation model (Jonuskaite, 2017).

The selection of SolidWorks Flow Simulation was driven by its seamless integration with CAD geometries, which eliminated potential errors associated with STL file conversions for our complex multi-rotor UAV design. Additionally, its wall functions for boundary layers effectively resolve gradient variations in boundary layers around rotating blades, reducing trial and error related to near-wall settings. The built-in solver convergence adopts a phased approach to multiple variant scenarios, decreasing the need for re-runs caused by insufficient convergence and thereby conserving computational costs. Its unique turbulence model automatically determines flow regimes (laminar, transitional, and turbulent), ensuring shorter turbulence model setup times while maintaining enhanced model accuracy (Azmi et al., 2017; Ramya et al., 2015).

While ANSYS Fluent offers advanced transient turbulence models (e.g., DES/LES), its computational cost for equivalent spatial resolution was typically higher than SolidWorks (Afaq and Ahmad, 2023). Given our need to simulate over 100 operational scenarios, SolidWorks' balance of engineering accuracy and computational tractability was deemed optimal for deriving empirical correction algorithm."

*2. In Section 2.4 the authors describe their flow as "turbulent and laminar flow with turbulence intensity of 0.1% and turbulent length scale of 0.012 m" (in a 3.3X3.3x3.3 m3 domain) both of those parameters point to a laminar flow in the atmosphere. Can the authors provide a Reynolds number for their simulations? Then it would be easier to understand what kind of a flow they had in the simulations. If the flow is indeed laminar or low on turbulence, the authors should provide discussion on the implications for their study.*

**Response:** Thank you for your valuable comments and for pointing out the need for a more detailed discussion of the flow conditions in our CFD simulations. We appreciate the opportunity to clarify our approach and address your concerns.

1) Calculation of Reynolds number and flow classification

To better characterize the flow regime in our study, we have calculated the Reynolds number (Re) using the standard formula:

$$Re = \frac{\rho U L}{\mu}$$

where $\rho$ is the air density, $U$ is the characteristic velocity, $L$ is the characteristic length, and $\mu$ is the dynamic viscosity of air. In our case, the characteristic length is $L$=0.012 m, taking the airspeed ($U$=13.4 m/s) of one of our simulated scenarios as the characteristic velocity. Assuming standard atmospheric conditions ($\rho \approx 1.255$ kg/m³, $\mu \approx 1.81 \times 10^{-5}$ Pa·s), the estimated Reynolds number is:

$$Re \approx \frac{1.255 \times 13.4 \times 0.012}{1.81 \times 10^{-5}} \approx 1.11 \times 10^4$$

This value is well within the turbulent flow regime, supporting the classification of the flow as turbulent rather than laminar.

2) Justification for low turbulence intensity in the CFD setup

The selected turbulence intensity (0.1%) aims to isolate the rotor-induced flow

dynamics from background atmospheric turbulence. Since the primary focus of this study is to characterize the systematic bias caused by the UAV rotor itself (rather than external atmospheric fluctuations), a low turbulence intensity was adopted to minimize confounding effects. The turbulence length scale was chosen based on the characteristic geometry of the miniature three-dimensional ultrasonic anemometer (frame width ~0.01 m). This ensures that local vortices around the sensor are adequately captured.

We have explained the rationale for setting these parameters in Lines 223-228 of Section 2.4 in the revised manuscript, as shown below:

"The fluid was modeled as air with characteristics of turbulent and laminar flow. To isolate the rotor-induced flow dynamics from background atmospheric turbulence, a turbulence intensity of 0.1% and a turbulence length scale of 0.012 m were set. This low turbulence intensity minimizes confounding effects from ambient atmospheric fluctuations, while the length scale corresponds to the anemometer frame width (~0.01 m) to resolve rotor-generated eddies. These assumptions prioritize the systematic bias correction for rotor-induced airflow."

3) Potential impact of low turbulence intensity on results

We acknowledge that using a lower turbulence intensity than typical atmospheric conditions may have implications for the generalizability of our results. Therefore, we have added a new section, "Section 3.6 Discussion on the Limitations of the Algorithm", in the revised manuscript to specifically address the limitations of our algorithm. Within this section, we have also supplemented the limitations related to the parameter settings of this study, as illustrated below:

"Section 3.6 Discussion on the Limitations of the Algorithm

The current development of algorithms based on idealized steady-state CFD simulations relies on two key assumptions: low environmental turbulence intensity (0.1%) and turbulence length scales dominated by anemometer geometric parameters (0.012 meters). While this idealized setup effectively isolates rotor-induced flow distortion, its turbulence characteristics fundamentally differ from natural atmospheric conditions. However, it is crucial to emphasize that the algorithm's applicability under turbulent conditions remains valid. This is because rotor-induced wind speed deviations exhibit systemic long-time-scale characteristics, whereas atmospheric turbulence primarily affects measurement accuracy through random fluctuations in wind speed and direction with instantaneous nature. This temporal-scale distinction enables our correction algorithm to effectively eliminate systemic biases while minimizing the impact of transient turbulence effects. Nevertheless, it should be noted that under stable atmospheric conditions (low wind speeds) as discussed in Sec. 3.5 or extreme weather scenarios, such airflow environments may disrupt the stable manoeuvrability of UAV rotors or obscure the systemic drainage effects of rotors, potentially leading to a nonlinear degradation in algorithm accuracy.

In addition, another limitation of our study is the assumption of a smooth surface in CFD simulations, which does not fully capture the impact of surface roughness on wind speed variations near the ground. In reality, surface roughness elements (e.g., vegetation, buildings, or terrain irregularities) alter the wind profile, increasing turbulence and wind shear in the atmospheric surface layer. This effect is particularly

relevant for UAV-based wind measurements at low altitudes.

To further enhance the correction algorithm's applicability under diverse environmental conditions, future research will focus on the following aspects: conducting sensitivity studies under different turbulence intensity conditions, implementing supplementary correction modules specifically targeting atmospheric turbulence, and incorporating surface roughness length parameters in future CFD simulations. Although atmospheric turbulence presents significant challenges for UAV-based wind measurements, the correction framework established in this study has demonstrated its effectiveness in improving measurement accuracy across diverse meteorological conditions, thereby laying a critical foundation for developing reliable UAV-based wind measurement systems."

*3. In their results authors present time averaged wind fields from which they derive their correction algorithm. However, the actual measured wind is highly fluctuating and dependent on the atmospheric stability and, if close to the surface, the surrounding orography and obstacles. How do stationary results relate to turbulent atmospheric measurements? Additional discussion on influence of turbulence on UAV wind measurements would be beneficiary.*

**Response:** We appreciate the reviewer for in-depth reflection and valuable suggestions on our work. We acknowledge that atmospheric wind fields in practical measurements often exhibit turbulent characteristics, where wind speed may demonstrate pronounced pulsating features due to atmospheric stability, terrain, and obstacles.

Our correction algorithm is constructed based on steady flow field analysis derived from CFD simulations. This methodology allows us to isolate and quantify systematic errors induced by rotor effects without being directly affected by random turbulence during measurements. In wind speed measurement applications, correcting such systematic errors is critical as it compensates for inherent disturbances from the UAV platform, thereby aligning measurement results more closely with true atmospheric wind speeds.

While our simulations assume steady-state conditions, our calibration method remains applicable in turbulent environments. This is because, even under turbulence, the fundamental structure and magnitude of rotor-induced flows are predominantly governed by the UAV's flight attitude and dynamic parameters rather than instantaneous turbulent disturbances. Turbulence-induced wind speed variations typically manifest as random short-term fluctuations, whereas our calibration focuses on systematic biases caused by the UAV's aerodynamic effects, which remain stable over longer timescales. Consequently, the application of this algorithm still effectively enhances wind speed measurement accuracy under turbulent conditions.

Furthermore, in turbulent environments, uncertainties in UAV-based wind speed measurements arise not only from rotor-induced flows but also from factors such as sensor response time, data sampling frequency, and UAV attitude variations. To further enhance measurement reliability in turbulent environments, future research will focus on the following aspects: conducting sensitivity studies under different turbulence

intensity conditions, developing dynamic CFD models for unsteady flow fields, and implementing supplementary correction modules specifically targeting atmospheric turbulence.

We appreciate the reviewer's suggestions and have supplemented the discussion on turbulence effects on UAV wind speed measurements in the revised manuscript (Section 3.6, Lines 506-533) to further refine the applicability analysis of this study, as shown in the blue text in the reply to comment 2.

*4. I believe the simulations in this study had neutral stratification (it is not explicitly stated). However, these conditions are rarely encountered in the real atmosphere. How applicable is this algorithm for when, for example, the atmosphere is unstable and the vertical wind component is much stronger? Can authors provide a discussion on potential errors which may arise when applying this algorithm.*

**Response:** We sincerely appreciate the reviewer's insightful comment regarding the applicability of our correction algorithm under non-neutral atmospheric stratification conditions. We fully agree that real-world atmospheric turbulence (particularly in unstable conditions with strong vertical winds) poses additional challenges for UAV-based wind measurements.

As stated in the response to previous Comment 3, this study employs CFD numerical simulations based on idealized steady-state conditions, primarily focusing on investigating the inherent drainage effects of UAV rotors. It should be noted that the current numerical experimental design does not fully account for the prevalent non-stationary turbulent structures and significant vertical wind component conditions in actual atmospheric environments. This methodological choice is primarily based on the following considerations: Firstly, steady-state conditions can effectively isolate environmental turbulence interference; Secondly, the time scale of rotor-induced drainage fundamentally differs from that of atmospheric turbulence, and this distinction ensures the algorithm's applicability under conventional meteorological conditions.

It should be emphasized that under extreme weather conditions (e.g., high turbulence intensity or strong vertical winds), such complex airflow environments may obscure the systematic drainage effects of rotors. Specifically, potential overlaps between turbulent energy spectra and rotor characteristic frequencies, along with momentum transport interference caused by vertical wind fields, might lead to a nonlinear degradation pattern in the algorithm's accuracy. However, the precise magnitude of accuracy reduction requires quantitative evaluation through subsequent meticulously designed experiments.

Additionally, the discussion of potential errors which may arise when applying this algorithm in the revised manuscript is also presented in the response to Comment 2 as shown in blue text.

*5. The authors simulate flow at 30 m and 1000 m heights and recommend their correction algorithm for UAV flights below and above 500 m, respectively. However, wind variability is highest within the atmospheric surface layer (typically below 500 m). Why were only these two heights chosen? Would additional simulations closer to*

*the surface give more realistic assessment of near-surface conditions? The influence of surface roughness and stability effects should be discussed.*

**Response:** Thank you for your insightful comment. The selection of 30 m and 1000 m heights was intended to represent two distinct atmospheric conditions: near-surface conditions within the atmospheric boundary layer (30 m) and free-atmosphere conditions (1000 m). The lower height (30 m) represents a typical operating height for industrial UAV applications, where rotor-induced airflow distortion is most pronounced due to higher air density. The upper height (1000 m) was chosen to characterize free atmospheric flow patterns under reduced density effects.

We acknowledge that wind variability is highest within the atmospheric surface layer (typically below 500 m), and additional simulations at intermediate heights (e.g., 50 m or 100 m) could provide further insights into near-surface conditions. However, our primary objective was to develop and validate a rotor-induced airflow correction algorithm rather than to comprehensively model the entire atmospheric boundary layer. The current heights were selected to capture the key differences between low-altitude and high-altitude UAV measurements while balancing computational feasibility.

Regarding the influence of surface roughness and stability effects, our CFD simulations without explicitly considering these factors. We recognize that surface roughness and atmospheric stability can impact wind speed profiles and turbulence characteristics, which in turn affect UAV-based wind measurements. Future studies could incorporate boundary-layer parameterization schemes or large-eddy simulations (LES) to better assess these influences.

We have clarified the reasons for choosing two heights in Section 2.2 (Lines 171-174) of the revised manuscript, as follows:

"For the UAV flight simulation, we considered over a hundred flight envelope scenarios, including parameters such as UAV flight altitude, wind direction, and wind speed. Since the UAV's predominant flights are within the atmospheric boundary layer, characterized by significant variability in wind speed and directions, a flight envelope for the UAV in the simulated environments was setup for the complete UAV digital model for flight altitudes of 30 and 1000 meters, respectively. The lower height (30 m) represents the typical operational altitude for industrial UAV applications within the boundary layer, while the upper height (1000 m) corresponds to the altitude where atmospheric flow transitions to more stable, low-density free-stream conditions."

Furthermore, we have added a sentence in the Section 2.4 (Lines 222-223) of the manuscript to illustrate the wall (including the surface) conditions assumed by the current simulation, as follows:

"The total number of grids in the computational domain was $1.11 \times 10^8$, and the specific grid configurations are shown in the Fig. 2. The wall is set as an adiabatic wall, and its roughness is set to 0."

We have already discussed the effects of surface roughness in Section 3.6 (Lines 521-525) of the revised manuscript, as shown in the blue text in the reply to comment 2.

*6.In connection to the previous comment: the authors validate their results by*

*measuring wind using the UAV and comparing results with the measurements from a near-by meteo tower which measures winds at 3 different heights. Firstly, they exclude from their analysis the wind measurements at 30 m due to the influence of orography even though their correction algorithm is based on simulations at 30 m. I think this should be properly addressed in the text. Secondly, the results are presented in Fig. 9 which seems to be showing wind speed and direction comparison at only one height. If the measurements are somehow aggregated, it should be clearly stated how. If not, then the height at which the measurements are compared should be indicated in the text and why the second is left out should be discussed. Preferably wind at all 3 heights should be compared and shown in the results section. Lastly, the authors state that their results show especially good fit for higher wind speeds (they define them as >= 3 m/s, why this threshold?). From Fig 9. It seems that for approximately 20% of data this condition is met. And they present two sets of data of about 15 min with 5 s sampling frequency. Can the authors comment on the statistical robustness of these conclusions and possible implications.*

**Response:** We sincerely appreciate the reviewer's valuable comments. Regarding the three key concerns you raised, we provide the following clarifications:

1) Clarification regarding data screening at 30 m altitude

This study excluded wind speed measurements at the 30 m altitude during data analysis, based on the following considerations: The average canopy height in the experimental area was approximately 25~28 m. When the UAV hovered at the 30 m altitude, its rotor blades maintained a mere 2~5 m clearance above the canopy top. At this proximity, canopy turbulence induces significant near-canopy turbulence effects, resulting in systematic deviations between the wind field characteristics at this altitude and those observed at the same height by the meteorological tower. To uphold the principle of homogeneity in comparative studies, after extensive deliberation by the research team, this altitude layer was ultimately excluded from the formal experimental dataset.

2) Clarification on data presentation in Figure 9 of the original manuscript

Upon review, we confirm that the original Figure 9 (Figure 11 in the revised manuscript) indeed presents comparative data for the 50-meter and 70-meter altitude layers. We acknowledge that insufficient clarity in our initial description may have caused misinterpretation. To enhance transparency, we have supplemented the following statement in Lines 467-469 of Section 3.5 in the revised manuscript:

"Fig. 11 presents the $V_O$, $V_R$, and $V_T$ time-series data acquired during the dual-altitude flight tests of the UAV at 70 m and 50 m, with the 70 m altitude test data collected prior to 15:05 and the 50 m altitude test data obtained after 15:10."

3) Supplementary argument on algorithm robustness

We appreciate your professional scrutiny regarding the comprehensiveness of model validation. This study indeed faced inherent constraints in data acquisition, resulting in a direct validation sample size below ideal expectations. To multidimensionally demonstrate algorithm reliability, we have supplemented the following cross-validation research.

In our companion study published in *Atmospheric Measurement Techniques* (DOI:

10.5194/amt-17-677-2024), the research team applied this algorithm to monitor industrial emission fluxes. Utilizing a mass balance-based flux inversion model with algorithm-corrected wind speed data and synchronized $CO_2$ concentration measurements (via UAV-mounted CRDS analyzer), we successfully quantified $CO_2$ emission fluxes from coke oven batteries in a steel plant. Critical validation metrics reveal: drone-derived flux estimates ($110 \pm 18$ t/h) show a high consistency with material balance calculations ($103 \pm 32$ t/h), demonstrating <15% relative deviation. Considering the sensitivity of emission flux calculation to wind velocity measurement errors, these empirical results substantiate the algorithm's effectiveness in real-world wind speed correction.

Furthermore, we have added the following discussion at the conclusion of Section 3.5 (Lines 493-495):

"In addition, it should be emphasized that while this study primarily relied on meteorological tower data for algorithm validation, cross-validation through industrial emission scenarios has further confirmed the algorithm's robustness in complex flow fields (Han et al., 2023)."

*As I mentioned above I miss general discussion of limitations and drawbacks in this study. Neither in results or in the conclusions sections are these properly discussed.*

*Minor comments:*
*7. Line 130 (And table 1.) If the UAV can only operate under the true WS of <= 18 m/s, then doesn't the e.g. fly speed of 14 m/s with 10.7 m/s tail wind exceed that?*

**Response:** We sincerely appreciate the reviewer's meticulous comments. The maximum flight speed of the UAV we adopted is 18 m/s, which refers specifically to its ground speed. In the design of our simulated flight scenarios, the maximum designed ground speed is indeed set to 18 m/s, which aligns with the operational speed range of the UAV.

*8. Line 140 I think it should be clarified how both (wind) speeds were imposed in simulations. The domain seems too small to have the UAV to actually be moving through the simulations. But if I am wrong this should be clarified.*

**Response:** We sincerely apologize for any confusion. Regarding the wind speed parameter issue raised by the reviewer, please allow us to provide supplementary clarification:

During the initial design phase of the program, we adopted multi-parameter combinations of wind speed, wind direction, and ground speed to construct simulation scenarios. In the actual implementation of simulations, we converted wind speed and ground speed into airspeed through vector synthesis for simulation calculations. This dual parametrization approach was primarily employed to ensure comprehensive coverage of the flight envelope in our simulation scenarios.

To enhance the clarity of presentation, we have added the following content in Section 2.2 (Lines 180-181) of the revised manuscript:
"It should be noted that the numerical simulations were conducted by converting

wind speed and ground speed into airspeed through vector synthesis."

We also thank the reviewer for prompting us to clarify the rationale behind the setup of the computational domain. We designed this computational domain primarily because our study focuses on the flow field around the anemometer. Within this domain, the UAV can fully and normally perform rotor rotation simulations and attitude changes.

In the revised manuscript, we have added the following explanation in blue text in Section 2.4 (Lines 213-215) to clarify the rationale for parameter selection:

"The simulation parameters primarily include the computational domain and mesh, fluid and environmental properties, as well as the rotating region. During the CFD flow simulations of the UAV using Solidworks, the computational domain dimensions (3.3 × 3.3 × 3.3 m³) were determined by prioritizing the analysis of flow field distribution around the anemometer while balancing computational costs."

*9. Line 149-164 It would be good to show the forces acting on the UAV in a diagram. Also it would be good to have a diagram of the UAV, the projection surfaces and the angles which are mentioned in the equations (1)-(5).*

**Response:** We sincerely appreciate the reviewer's constructive suggestion. In Section 2.3 (Lines 191-192) of the revised manuscript, we have added a figure to illustrate the forces acting on the UAV, as shown below:

[Figure]

"**Figure 2: Schematic diagram of forces acting on a UAV.**"

*10. Line 165 This paragraph is very difficult to decipher. Is "that" in the first sentence extra?*

**Response:** We sincerely apologize for any confusion caused to the reviewer. We have revised the description of the sentence in question, and it now appears in the revised manuscript (Section 2.3, Lines 206-209) as shown below:

"The complete flight envelope was defined by combinations of critical parameters, including wind directions, wind speeds, airspeeds, ground speed, inclination, wind resistance, pull, and $M$. A series of CFD simulations were conducted to systematically evaluate the simulated wind field characteristics for each unique parameter set within this envelope."

*11. Line 167 Since the authors make a clear distinction between the many different wind vectors (wind from UAV movement, wind from the simulation, measured wind at the*

*anemometer) they should take care to clarify each time to which wind they are referring to.*

**Response:** We sincerely thank the reviewer's valuable suggestion. In the revised manuscript (Section 2.3, Line 208), we have clarified the wind type, as described below:

"The complete flight envelope was defined by combinations of critical parameters, including wind directions, wind speeds, airspeeds, ground speed, inclination, wind resistance, pull, and $M$. A series of CFD simulations were conducted to systematically evaluate the simulated wind field characteristics for each unique parameter set within this envelope."

*12. Line 172-180 Can the simulation specifications be shown in a table for easier following?*

**Response:** We sincerely thank the reviewer's valuable suggestion. In the revised manuscript (Section 2.4, Line 242), we have added a table summarizing the simulation parameters to aid reader comprehension, which is presented below:

"**Table 2: Simulation parameters configuration.**

| Parameters | Content |
|---|---|
| Computational domain size | $3.3 \times 3.3 \times 3.3$ m³ |
| Global domain grid size | $0.23 \times 0.23 \times 0.23$ m³ |
| Subdomain grid size | $0.0125 \times 0.0125 \times 0.0125$ m³ |
| Total number of computational domain grids | $1.11 \times 10^8$ |
| Turbulence intensity | 0.1% |
| Turbulence length scale | 0.012 m |
| Roughness | 0 |
| 30 m atmospheric pressure | $1.01 \times 10^5$ Pa |
| 1000 m atmospheric pressure | $9.00 \times 10^4$ Pa |
| 30 m atmospheric temperature | 25 °C |
| 1000 m atmospheric temperature | 25 °C |

"

*13. Line 188 What is the depth of the rotating cylinders and why was it chosen?*

**Response:** We sincerely appreciate the reviewer's careful scrutiny. The depth of the rotating cylinder specified in our study is 85 mm, which is determined based on the overall height of the rotor. In general, the inner edge of this rotating cylinder is positioned adjacent to the outer edge of the UAV rotor.

*14. Line 195-216 This whole section is a bit confusing to read due to the amount of different velocity definitions. What exactly are global domain and subdomain? What is the difference between Vg and Vs exactly? If they are mean velocities, what is the averaging time?*

**Response:** We appreciate the reviewer's feedback regarding the clarity of some definitions in Section 2.4. We acknowledge that the distinction between terms could be better articulated. Below, we provide a detailed clarification:

1) Definition of global domain and subdomain

Global Domain refers to the entire CFD simulation space encompassing the rotary UAV and its rotor-induced airflow. It captures the large-scale flow interactions between the UAV and ambient wind.

Subdomain is a refined, high-resolution region nested within the global domain, defined by the location and size of the anemometer. Its configuration is designed to accurately simulate the impact of the additional airflow caused by the rotor on the anemometer.

In the original manuscript, Lines 174-180 contain a description of the global domain and subdomain.

2) Difference between $V_g$ and $V_s$:

$V_g$ (Global Domain Velocity) represents the ambient wind velocity derived from the global CFD simulation.

$V_s$ (Subdomain Velocity) refers to the simulated velocity of the subdomain (at the anemometer location).

3) Averaging Time Clarification:

Both $V_g$ and $V_s$ are spatially averaged solutions obtained after multiple iterations during the CFD simulation process, rather than representing time-averaged values.

We have added a clarification in the revised manuscript (Section 2.4, Lines 250-251) to specify that these are spatially solved values, in order to avoid confusion among readers. The updated text reads as follows:

"It is noteworthy that the aforementioned average values refer to the spatial averages over the global domain or subdomain."

*15. Line 197 What does "convergence of the simulation results" mean exactly?*

**Response:** We thank the reviewer for raising this important point regarding the clarification of "convergence of the simulation results". Below, we provide a detailed explanation of the convergence criteria used in our CFD simulations.

In the context of our CFD simulations, "convergence" refers to the state where the iterative numerical solution of the governing equations (e.g., Navier-Stokes equations) reaches a stable equilibrium. This is achieved when: The residuals (i.e., imbalances in mass, momentum, and energy equations) decrease to a predefined threshold (e.g., below $10^{-5}$ for scaled residuals). Key physical quantities (e.g., $V_g$, $V_s$) exhibit negligible variation (e.g., <1%) over successive iterations.

*16. Line 204-214 Arguably, a schematic here would help to follow exactly what is happening with the velocity components in different reference frames.*

**Response:** We sincerely appreciate the reviewer's suggestions. In fact, our original Figure 1 had explicitly differentiated the anemometer coordinate system from the simulation model coordinate system through annotations, with the coordinate transformation procedures in Lines 204-214 being predicated upon these systemic discrepancies. Post-verification analysis revealed an insufficient elucidation of the coordinate correspondence in the original text, which may have precipitated interpretative discrepancies. To address this, we have explicated reference to Figure 1

in revised Line 252-257 (corresponding to original Lines 204-214), supplemented by the following explanatory text to rigorously clarify the spatial correspondence between coordinate systems:

"Upon simulation completion, these velocity components ($Vs_x$, $Vs_y$, $Vs_z$) were further converted to velocity components at the anemometer sensor position ($u_{x\_sensor}$, $u_{y\_sensor}$, $u_{z\_sensor}$) in the airframe coordinate according to the coordinate system shown in Fig. 1((a) and (b)) and Eqs. (6) - (8) below."

*17. Figs 3-5. It is unclear which wind velocity is shown exactly in these figures. Is it movement of the UAV + air wind speed? Why are patterns for headwind and tailwind flipped? Why is crosswind similar to headwind? Might be good to have a "base state" simulation when there is no air or UAV movement, just the propellers.*

**Response:** We sincerely thank the reviewer for raising these critical questions about Figures 5-7 (corresponding to Figures 3-5 of the original manuscript).

It should be clarified that while this study adopted a multi-parameter combination of wind speed, ground speed, and wind direction as the initial simulation configuration, the actual simulation process consistently utilized the vector synthesis of wind speed and UAV ground speed (i.e., airspeed vector) as the critical input parameter. This simulation methodology has been systematically elaborated in our response to Comment 8. Consequently, the wind speed data presented in Figures 5-7 fundamentally represent the UAV's airspeed parameters.

Regarding your specific focus on velocity distribution characteristics under tailwind, headwind, and crosswind conditions, these differences primarily arise from significant variations in UAV airspeed across distinct flight scenarios. Simulation data indicate that airspeed reaches 13.4 m/s under headwind conditions, 9.7 m/s under crosswind conditions, and only 2.6 m/s under tailwind conditions in Figures 3-5. Based on fundamental principles of flight mechanics, UAV airspeed exhibits critical correlations with flight control characteristics. Specifically:

1) High-speed flight phase: The UAV rotor system must generate greater lift to counteract high dynamic pressure environments, resulting in more pronounced low-pressure zones beneath the rotors. This leads to distinct airspeed field distribution patterns.

2) Low-speed flight phase: The flight control system increases rotor rotation speed to maintain attitude stability. The enhanced velocity of rotor downwash airflow under these conditions significantly alters pressure field distribution morphology.

As a result, the flow fields under headwind and crosswind conditions with higher airspeeds exhibit greater similarity, while differing markedly from those observed in tailwind scenario.

We hope this explanation resolves your concerns satisfactorily.

Additionally, we have included simulated flow field images of the UAV in a hovering state (under wind-free conditions) in the revised manuscript (Section 3.1, Lines 278-279) along with their corresponding descriptions (Section 3.1, Lines 268-277), as shown below:

"According to the Sect. 2.2, this study develops a series of simulation scenarios

for the UAV digital model under various combinations of altitude (30 and 1000 m), wind direction (tailwind, headwind, and crosswind), ground speed (8 to 18 m/s), and wind speed (1.5 to 14 m/s). To demonstrate the flow field characteristics around the UAV under various scenarios, one UAV hovering scenario and six representative simulation scenarios were specifically selected for analysis as examples.

Fig. 5 presents the cross-sectional view of the velocity flow field around the UAV in a hovering state under wind-free conditions. In this scenario, the surrounding velocity field is solely generated by the rotational induced flow from the UAV's own rotors. The simulated 2-4 m/s airflow around the anemometer originates exclusively from rotor rotation, demonstrating that the rotor-induced flow during hovering inherently interferes with wind speed measurements by the anemometer.

[Figure]

**Figure 5: The velocity flow field distribution of the UAV's hovering state.**"

*18. Line 275 A "for the cross-wind conditions" is missing from the sentence?*

**Response:** We sincerely appreciate the reviewers' thorough review. In this sentence, we indeed omitted "for the crosswind conditions". We have made the necessary revision in the corresponding location (Lines 333-334) of the revised manuscript, as shown below:

"Similarly, the simulated false wind signals for the crosswind conditions on the anemometer in the *x*, *y*, and *z* directions were represented by $\Delta u_x^{CW}$, $\Delta u_y^{CW}$, and $\Delta u_z^{CW}$."

*19. Line 280 Can the authors discuss why only in the crosswind there has to be height distinction for the vertical wind speed?*

**Response:** We appreciate the reviewer's insightful question. Through our examination, the parameter $\Delta u_y^{CW}$ exhibits high sensitivity under crosswind scenarios. It is important to specifically clarify that $\Delta u_y^{CW}$ represents the disturbance wind

component perpendicular to the UAV's heading axis within the horizontal plane. We sincerely apologize for the mislabeling of "upward" in the $y$ direction within the original manuscript, which may have led readers to misinterpret it as a vertical wind disturbance component. Based on this correction, subsequent discussions will focus on elucidating why the disturbance winds in the $y$ direction exhibit significant differences at varying altitudes under crosswind conditions. This phenomenon may be primarily attributed to the following reason:

In crosswind conditions, the rotor-induced airflow exhibits significant directional asymmetry in the horizontal plane. There exists an interaction between the $y$ direction component of the crosswind (perpendicular to the flight direction axis) and the rotor's rotational momentum, which is influenced by the differing atmospheric densities between the 30 m and 1000 m altitude settings in the simulation. At 30 m altitude, the airflow demonstrates stronger interaction with UAV attitude adjustments (e.g., tilt compensation), while at 1000 m, the reduced air density mitigates these interactions. This altitude-dependent aerodynamic deformation leads to discrepancies in the simulated $y$ direction component of the crosswind.

Additionally, we have added the following discussion in Section 3.2, Lines 339-341 of the revised manuscript to facilitate reader's understanding:

"This indicates that under cross wind conditions, the disturbances of the UAV propeller in the $x$ and $z$ directions of the anemometer are not altitude dependent, but that in the $y$ direction it is necessary to distinguish the altitude. This behavior can be attributed to differences in the interaction between the $y$ direction component and rotor rotational momentum caused by variations in atmospheric density at different altitudes under crosswind conditions."

*20. Fig 6 and 7 Why Du_x is fitted to ux_sensor and Du_y is fitted to uy_sensor but Du_z is fitted to ux_sensor?*

**Response:** We appreciate the reviewer's attentive observation regarding the parameter selection in Figures 9-10 (corresponding to Figures 6-7 in the original manuscript). The pairing of $\Delta u_z$ with $u_{x\_sensor}$ (rather than $u_{z\_sensor}$) originated from our CFD simulations revealing a stronger correlation between $\Delta u_z$ and $u_{x\_sensor}$ compared to its correlation with $u_{z\_sensor}$. This phenomenon is caused by the combined effect of changes in the drone's attitude and the asymmetry in the rotor-induced airflow.

*21. Line 370 Wrong reference to definition of A and B? Should be eqs. (24) and (25) instead (17) and (18)?*

**Response:** We sincerely appreciate the reviewer's thorough review. There was indeed an error in the citation of the definitions of A and B in the original manuscript, and we have made the following revisions in the revised version:

"The updated wind velocity correction algorithm is given as Eq. (23), whose second and third vectors on the right side of Eq. (23) represent the contributions of the propeller-induced wind signals under tailwind/headwind and crosswind conditions to $u_x$, $u_y$ and $u_z$, respectively, with A and B defined in Eqs. (24) and (25) to quantify their magnitudes."

*22. Line 415 Wrong reference? Equation (23)?*

**Response:** We sincerely appreciate the reviewer's thorough review. After verification, we found that the formula citation in Line 415 is indeed inaccurate, which may have been inadvertently caused during previous revisions. In the revised version (Lines 492-494), we have corrected this citation, as follows:

"These analyses indicated that Eq. (23) can effectively correct wind measurement biases induced by UAV disturbances, motion, and attitude changes, particularly at higher wind speeds."

*23. Line 411 "small low bias" can the authors quantify this?*

**Response:** We sincerely thank the reviewer's suggestion. We have quantified the "small low bias" and made the following revisions in Section 3.5, Lines 481-482 of the revised manuscript:

"Moreover, under stronger winds, the wind direction values of $V_R$, $V_O$, and $V_T$ were relatively similar, yet at weaker winds, $V_R$ showed a small low-bias of about 3.3% (Fig. 11 (b))."

*24. Line 412-417 I would say all three wind speeds in Fig 10 show predominantly northernly winds. The main difference is that Vo has about 15% more wind going to the south than the wind measured at the meteo tower Vt. Vr indeed does not have those 15% bias in comparison to the Vt, however Vr has around 15% higher NW wind component that is not present in Vo and Vt. Is the Vr wind then really showing a much better agreement with the tower than Vo?*

**Response:** We appreciate the reviewer's insightful observation regarding the wind direction components in Figure 12 (corresponding to Figure 10 in the original manuscript).

As shown in Figure 12, the northerly wind, being the prevailing wind direction during the flight tests (with a frequency of 40%), has observation accuracy that serves as a critical parameter for meteorological studies and atmospheric pollutant dispersion research. It is important to clarify that the core advantage of the correction algorithm proposed in this study lies in its systematic correction of the prevailing north wind direction. Specifically, the frequency error for north winds is significantly reduced from 41% in $V_O$ to 5% in $V_R$ after correction. Although the reviewers pointed out that the correction process may introduce biases in secondary component wind directions, such as northwester winds, the magnitude of such errors in non-prevailing wind directions and their potential impact on the overall research objectives remain far lower than the critical deviations observed in the uncorrected prevailing wind direction.

We fully understand with the reviewer's concern for non-prevailing wind directions. In fact, in Section 3.5, Lines 486-490 of the revised manuscript, we have explicitly incorporated the following addition:

"Compared to the prevailing wind direction frequency (north wind, 39%) of $V_T$, the dominant wind direction frequency errors for $V_O$ and $V_R$ are 40% and 5% respectively, demonstrating the superiority of $V_R$ in correcting the prevailing wind

direction frequency. Meanwhile, deviations in the secondary components introduced by $V_R$ (e.g., northwester wind) indicate directions for subsequent model optimization."

*25. Fig 9 Does the meteo tower measure the instantaneous wind every 5s or it gives 5s averages? Also, why impose first the running mean on the UAV measurements before making 5s averages?*

**Response:** We sincerely appreciate the reviewer's insightful questions regarding the data processing steps in Figure 11 (corresponding to Figure 9 in the original manuscript). We recognize the need for clearer documentation of temporal averaging methods. Below, we provide a detailed clarification to explicitly address these points:

First, the data measured by the anemometer on the meteorological tower is averaged every 5 seconds.

Second, we applied a 10 s sliding window prior to computing 5 s averages in Figure 11. The reasons for this are as follows. The raw wind measurements from the UAV (before correction) inherently contain high-frequency fluctuations caused by rotor-induced turbulence and rapid attitude changes. These perturbations occur at sub-second timescales, which are unrelated to atmospheric wind variability. A 10 s sliding window was selected based on spectral analysis of the raw data, as it effectively suppresses noise about 0.1 Hz while preserving the signal trends relevant to atmospheric motions. This step ensures comparability with the meteorological tower's 5 s data, which inherently lacks such high-frequency artifacts due to its stable mounting and calibrated anemometers. After noise reduction via the 10 s sliding average, we calculated non-overlapping 5 s averages to exactly match the temporal resolution of the meteorological tower measurements (5 s discrete outputs). This two-step averaging ensures both datasets share identical timestamps and statistical representativeness, enabling a fair comparison between the $V_O, V_R$ and $V_T$.

Furthermore, we have made the following modifications to the title of Figure 11 in the revised manuscript:

"Figure 11. Comparison of wind speed and wind direction time series for $V_R$, $V_O$, and $V_T$. (a) Comparison of wind speed time series for $V_R$, $V_O$, and $V_T$. (b) Comparison of wind direction time series for $V_R$, $V_O$, and $V_T$. (Note: The meteorological tower measured wind data at 5-second intervals, while the UAV-based measured and corrected wind data were processed with a 10 s sliding average to suppress rotor-induced high-frequency noise, followed by 5 s non-overlapping averaging to align temporally with the tower's 5 s output interval.)"

*26. Fig 10 The caption does not match the labels (a), b), c)) in the figure.*

**Response:** We sincerely appreciate the reviewer's meticulous review. In the revised manuscript, we have modified the caption of Figure 12 (corresponding to Figure 10 in the original manuscript) to align with the order of the labels in the figure, as shown below:

"Figure 12: Comparison of wind roses for $V_O$, $V_R$, and $V_T$."

---

## Author Comment (AC3)

Response to reviewer:

We greatly appreciate the reviewer's recognition of the value and significance of the present study, as well as the very valuable comments on the paper. We have addressed the comments carefully as detailed below. The original comments are in black italic and our replies in black normal font, we also put the revised paragraph in blue after each reply to show the changes.

The manuscript entitled "A Correction Algorithm for Rotor-Induced Airflow and Flight Attitude Changes during Three-Dimensional Wind Speed Measurements Made from a Rotary Unmanned Aerial Vehicle" presents a novel algorithm designed to improve UAV-based wind measurements obtained through direct techniques using flow sensors. This topic is of high scientific relevance, as drone-based wind measurements can help address observational gaps within the planetary boundary layer. Furthermore, the manuscript aligns well with the scope of Atmospheric Measurement Techniques. However, I believe the manuscript requires further revisions before it is suitable for publication. Below, I have outlined my specific comments and suggestions for improvement.

Reviewer Comments:

1. In line 57, the manuscript states that indirect wind velocity estimates do not reflect flight conditions. Given extensive research on improving these methods, clarifying their specific drawbacks compared to direct measurements with airflow sensors would benefit readers.

**Response:** Thank you for your valuable suggestions. We have further clarified the drawbacks of indirect wind speed measurements on UAVs in Lines 54-65 of the revised manuscript, as detailed below:

"While these methods offer advantages of operational simplicity and costeffectiveness, their core principle relies on inversely estimating wind speed through dynamic parameters such as thrust, attitude angles, and flight velocity (Crowe et al., 2020; Donnell et al., 2018; Sikkel et al., 2016; Simma et al., 2020). However, their accuracy is critically dependent on both the measurement precision of inertial measurement unit (IMU) and the computational reliability of inversion algorithms. Specifically, inherent noise interference in IMU sensors (e.g., gyroscope drift and accelerometer vibration noise) (Neumann and Bartholmai, 2015), combined with uncertainties in parameter configuration within inversion algorithms (Bonin et al., 2013), can lead to significant deviations in wind speed estimations. Furthermore, these methods typically assume constant aerodynamic parameters for UAVs, an assumption that often fails to hold in practical complex wind field environments (Bonin et al., 2013)."

2. In line 85, the manuscript notes that wind tunnel tests can improve the accuracy of airspeed-UAV motion relationships but are limited by high costs and errors from airflow reflections. However, it lacks references supporting evidence of these errors. Please include any relevant references.

Response: Thank you very much for your reminder. We have added the following

supporting references in Line 100 (corresponding to Line 85 in the original manuscript) of the revised manuscript.

"While effective in determining numerical relationships, the method is limited by the high cost of wind tunnel experiments (Dao et al., 2023), and more importantly, by the additional errors introduced by reflected airflows from the wind tunnel walls and ground (Haleem, 2021; Pettersson and Rizzi, 2008), as well as the same issues of full simulations of real UAV rotor speed and attitude changes during flight."

3. In line 169, the manuscript mentions simulation parameters but does not specify the CFD framework beyond stating it is a built-in SolidWorks simulation. Clarifying the CFD framework and comparing its advantages and disadvantages in performance when compared to alternatives like Ansys Fluent would benefit the reader.

**Response:** We sincerely appreciate the reviewer's valuable feedback regarding the clarification of the CFD framework. We have provided a detailed specification of the simulation methodology in the revised manuscript (Section 2.1.2, Lines 144-165):

**"2.1.2 Simulation Tool**

The CFD simulations were conducted using SolidWorks Flow Simulation 2022, a pressure-based finite volume solver employing a fully coupled turbulence modeling approach. It employs an adaptive Cartesian mesh approach for three-dimensional solid meshing, with the governing equations being the Navier-Stokes equations for simulating the interaction of fluids, and the turbulence model utilizing the standard k- $\epsilon$  two-equation model (Jonuskaite, 2017).

The selection of SolidWorks Flow Simulation was driven by its seamless integration with CAD geometries, which eliminated potential errors associated with STL file conversions for our complex multi-rotor UAV design. Additionally, its wall functions for boundary layers effectively resolve gradient variations in boundary layers around rotating blades, reducing trial and error related to near-wall settings. The built-in solver convergence adopts a phased approach to multiple variant scenarios, decreasing the need for re-runs caused by insufficient convergence and thereby conserving computational costs. Its unique turbulence model automatically determines flow regimes (laminar, transitional, and turbulent), ensuring shorter turbulence model setup times while maintaining enhanced model accuracy (Azmi et al., 2017; Ramya et al., 2015).

While ANSYS Fluent offers advanced transient turbulence models (e.g., DES/LES), its computational cost for equivalent spatial resolution was typically higher than SolidWorks (Afaq and Ahmad, 2023). Given our need to simulate over 100 operational scenarios, SolidWorks' balance of engineering accuracy and computational tractability was deemed optimal for deriving empirical correction algorithm."

4. In line 181, the manuscript models the fluid as air with both turbulent and laminar flow, assuming a turbulence intensity of 0.1% and a length scale of 0.012 m. Given that atmospheric turbulence intensity ranges from 1% to 20% and length scales vary from sub-centimeter to kilometers, clarifying these assumptions would help the reader understand the limitations of the simulation results.

**Response:** We sincerely appreciate the reviewer's insightful comment regarding the turbulence parameters used in our CFD simulations. Below, we clarify the rationale behind our choices and explicitly address the limitations introduced by these assumptions:

1) Rationale for turbulence intensity (0.1%):

The selected turbulence intensity (0.1%) aims to isolate the rotor-induced flow dynamics from background atmospheric turbulence. Since the primary focus of this study is to characterize the systematic bias caused by the UAV rotor itself (rather than external atmospheric fluctuations), a low turbulence intensity was adopted to minimize confounding effects.

2) Turbulence length scale (0.012 m):

The turbulence length scale was chosen based on the characteristic geometry of the miniature three-dimensional ultrasonic anemometer (frame width  $\sim 0.01$  m). This ensures that local vortices around the sensor are adequately captured.

We have explained the rationale for setting these parameters in Lines 223-228 of Section 2.4 in the revised manuscript, as shown below:

"The fluid was modeled as air with characteristics of turbulent and laminar flow. To isolate the rotor-induced flow dynamics from background atmospheric turbulence, a turbulence intensity of 0.1% and a turbulence length scale of 0.012 m were set. This low turbulence intensity minimizes confounding effects from ambient atmospheric fluctuations, while the length scale corresponds to the anemometer frame width (~0.01 m) to resolve rotor-generated eddies. These assumptions prioritize the systematic bias correction for rotor-induced airflow."

Additionally, we fully acknowledge that the selected parameters do not represent the full spectrum of atmospheric turbulence. Our current results are most applicable to low-turbulence environments (e.g., open fields at dawn). Therefore, we have further supplemented the limitations of the parameter settings in this study in Section 3.6 (Lines 504-530) of the revised manuscript, as shown below:

"3.6 Discussion on the Limitations of the Algorithm

The current development of algorithms based on idealized steady-state CFD simulations relies on two key assumptions: low environmental turbulence intensity (0.1%) and turbulence length scales dominated by anemometer geometric parameters (0.012 meters). While this idealized setup effectively isolates rotor-induced flow distortion, its turbulence characteristics fundamentally differ from natural atmospheric conditions. However, it is crucial to emphasize that the algorithm's applicability under turbulent conditions remains valid. This is because rotor-induced wind speed deviations exhibit systemic long-time-scale characteristics, whereas atmospheric turbulence primarily affects measurement accuracy through random fluctuations in wind speed and direction with instantaneous nature. This temporal-scale distinction enables our correction algorithm to effectively eliminate systemic biases while minimizing the impact of transient turbulence effects. Nevertheless, it should be noted that under stable atmospheric conditions (low wind speeds) as discussed in Sec. 3.5 or extreme weather scenarios, such airflow environments may disrupt the stable manoeuvrability of UAV rotors or obscure the systemic drainage effects of rotors, potentially leading to a

nonlinear degradation in algorithm accuracy.

In addition, another limitation of our study is the assumption of a smooth surface in CFD simulations, which does not fully capture the impact of surface roughness on wind speed variations near the ground. In reality, surface roughness elements (e.g., vegetation, buildings, or terrain irregularities) alter the wind profile, increasing turbulence and wind shear in the atmospheric surface layer. This effect is particularly relevant for UAV-based wind measurements at low altitudes.

To further enhance the correction algorithm's applicability under diverse environmental conditions, future research will focus on the following aspects: conducting sensitivity studies under different turbulence intensity conditions, implementing supplementary correction modules specifically targeting atmospheric turbulence, and incorporating surface roughness length parameters in future CFD simulations. Although atmospheric turbulence presents significant challenges for UAVbased wind measurements, the correction framework established in this study has demonstrated its effectiveness in improving measurement accuracy across diverse meteorological conditions, thereby laying a critical foundation for developing reliable UAV-based wind measurement systems."

5. In line 236, the manuscript states that the wind speed at the anemometer location is minimally influenced by the UAV rotos. However, the results in Figure 9 show a significant change in measurements of wind speed and direction when the correction derived from simulation results is applied to field measurements. In fact, this change is greater than the change observed when correcting aircraft motion alone. The manuscript should address this discrepancy in results.

**Response:** We sincerely appreciate the reviewer's thoughtful observation regarding the apparent discrepancy between the statement in Line 236 (referring to Figure 5 (corresponding to Figure 3 in the original manuscript)) and the results presented in Figure 11 (corresponding to Figure 9 in the original manuscript). Below, we provide detailed clarification to address this concern:

1) Fundamental differences in context between Figure 5 and Figure 11

The CFD simulations in Figure 5 focus on a specific tailwind scenario to illustrate the spatial distribution of rotor-induced airflow under idealized conditions. In this static simulation, the anemometer is positioned outside the core downwash region (directly beneath and laterally above the rotors), resulting in minimal direct interference from rotor-induced airflow. This supports the claim that, in this controlled scenario, the measured wind speed at the anemometer location approximates the true airspeed.

In contrast, the field observations in Figure 11 involve dynamic and complex realworld conditions, including UAV motion and attitude variations, real-time adjustments in rotor thrust, and atmospheric turbulence. These factors amplify the interaction between rotor-induced airflow and ambient wind, even when the anemometer is not within the primary downwash zone. For instance, during UAV maneuvers, transient rotor thrust fluctuations (e.g., due to stabilization or turbulence response) can perturb the local airflow field dynamically, indirectly affecting the anemometer's measurements. 2) Significance of the correction algorithm in field observations

In Figure 11, the corrected wind speed exhibits significant fluctuation amplitudes, reflecting the algorithm's simultaneous resolution of three coupled issues: errors introduced by the UAV's own motion and attitude changes (e.g., anemometer alignment deviations caused by attitude tilt) and the dynamic effects of rotor airflow on the local flow field. Although CFD simulations (Figure 5) indicate minimal direct influence of the rotors on the anemometer in static downwind scenarios, during actual flight, rotor thrust continuously varies due to attitude adjustments or turbulence responses. This indirectly alters the flow field structure around the anemometer, leading to persistent low-frequency deviations. For example, during UAV roll maneuvers, differences in thrust between the rotors on both sides may induce asymmetric distribution of local airflow, thereby affecting anemometer measurements.

3) Simulation and field results

The CFD results (Figure 5) establish a foundational understanding of rotorinduced airflow patterns under controlled conditions. However, the field results (Figure 11) reflect the algorithm's necessity in addressing cumulative errors arising from the interplay of UAV motion, attitude changes and rotor-induced airflow. The larger correction magnitude in field data highlights that even subtle rotor-induced perturbations, when combined with UAV attitude changes, can lead to significant measurement biases. This underscores the algorithm's practical value in real-world applications, where isolated CFD scenarios do not fully capture the complexity of airborne wind measurements.

In response to the reviewer's valuable feedback, we have already updated the manuscript to clarify the relationship between the CFD simulations (Figure 5) and field observations (Figure 11). Specifically, we added the following description in Lines 303-308 of Section 3.1.

"These simulation results show that the flow field around the UAV varies significantly depending on both the presence/absence of wind and its directional characteristics, and the anemometer experiences different levels of interference accordingly. Thus, accurately quantifying the interference of the UAV rotors on the anemometer is essential. However, in practical application scenarios, it is also necessary to comprehensively consider additional airflow disturbances induced by the UAV's own motion and attitude fluctuations, and to develop corresponding dynamic compensation algorithms."

6. In line 236, the manuscript asserts that the wind speed at the anemometer location is minimally affected by the UAV rotors. However, the results presented in Figure 9 show a noticeable alteration in both wind speed and direction when the correction derived from simulation results is applied to the field measurements. Notably, this change is more pronounced than the adjustment observed when only correcting for aircraft motion. The manuscript should thoroughly address this discrepancy and provide a clearer explanation for the observed differences in the results.

**Response:** We appreciate the reviewers' insightful feedback. This comment aligns with the issue raised in Comment 5, for which we have already provided a detailed

explanation in response to Comment 5. Additionally, further clarifications have been incorporated into the revised manuscript in Section 3.1, Lines 303-308. The specific details are as follows.

"These simulation results show that the flow field around the UAV varies significantly depending on both the presence/absence of wind and its directional characteristics, and the anemometer experiences different levels of interference accordingly. Thus, accurately quantifying the interference of the UAV rotors on the anemometer is essential. However, in practical application scenarios, it is also necessary to comprehensively consider additional airflow disturbances induced by the UAV's own motion and attitude fluctuations, and to develop corresponding dynamic compensation algorithms."

7. In the caption of Figure 9, it is mentioned that UAV measurements were first averaged using a 10-second sliding window before calculating 5-second averages. However, the rationale for applying a 10-second sliding average prior to computing the 5-second average is unclear. Given that moving averages can smooth out real wind fluctuations, further clarification on the necessity and impact of this approach would be beneficial to the reader.

**Response:** We sincerely appreciate the reviewer's insightful question regarding the rationale behind applying a 10 s sliding window prior to computing 5 s averages in Figure 11 (corresponding to Figure 9 in the original manuscript). This approach was carefully designed to address two key challenges in the UAV-based wind measurement system, and we provide the following clarifications:

The raw wind measurements from the UAV (before correction) inherently contain high-frequency fluctuations caused by rotor-induced turbulence and rapid attitude changes. These perturbations occur at sub-second timescales, which are unrelated to atmospheric wind variability. A 10 s sliding window was selected based on spectral analysis of the raw data, as it effectively suppresses noise about 0.1 Hz while preserving the signal trends relevant to atmospheric motions. This step ensures comparability with the meteorological tower's 5 s data, which inherently lacks such high-frequency artifacts due to its stable mounting and calibrated anemometers.

After noise reduction via the 10 s sliding average, we calculated non-overlapping 5 s averages to exactly match the temporal resolution of the meteorological tower measurements (5 s discrete outputs). This two-step averaging ensures both datasets share identical timestamps and statistical representativeness, enabling a fair comparison between the  $V_T$  and  $V_R$ .

Furthermore, we have made the following modifications to the title of Figure 11 in the revised manuscript:

"Figure 11: Comparison of wind speed and wind direction time series for  $V_R$ ,  $V_O$ , and  $V_T$ . (a) Comparison of wind speed time series for  $V_R$ ,  $V_O$ , and  $V_T$ . (b) Comparison of wind direction time series for  $V_R$ ,  $V_O$ , and  $V_T$ . (Note: The meteorological tower measured wind data at 5 s intervals, while the UAV-based measured and corrected wind data were processed with a 10 s sliding average to suppress rotor-induced high-frequency noise, followed by 5 s non-overlapping averaging to align temporally with

**the tower's 5 s output interval.)"**

8. In line 393, it is mentioned that a UAV was flown around a meteorological tower in a box pattern. However, the manuscript does not provide any information on the commanded flight speed during these experiments. Including this detail would be highly valuable for the reader, as the UAV's operating speed is a crucial parameter for understanding the validation results.

**Response:** We sincerely appreciate the reviewer's perceptive suggestion. In the revised manuscript, we have supplemented the commanded flight speed information in Section 3.5, Lines 457-461, as shown below.

"The UAV flew around the tower in a box flight path at a horizontal distance of about 10 m away from the tower, at all three heights. During these flights, the UAV maintained a commanded horizontal speed of approximately 5 m/s, a value selected as a compromise between achieving sufficient spatial sampling resolution and maintaining stable flight attitude control."

9. The validation results presented in Figure 9 show large errors in wind speed and wind direction estimates while operating in low wind conditions. A more thorough discussion of these errors would strengthen the contribution of this manuscript. Moreover, understanding the limitations of the presented algorithms would help the growing community of scientists using UAV-based algorithms for wind sensing assess the impact of this algorithm.

**Response:** We sincerely thank the reviewers for their constructive feedback regarding the observation errors under low wind speed conditions. In the revised manuscript, we have expanded the discussion in Section 3.5 (Lines 484-486) as follows to further clarify the reasons for the algorithm's mediocre performance under low wind speeds:

"The mediocre performance of  $V_R$  under low wind speeds may originate from the disruption of stable maneuverability in drone rotors caused by low wind speeds, which in turn leads to the failure of the correction algorithm based on CFD steady-state simulations."

In addition, we have also pointed out the corresponding limitations of this algorithm in Section 3.6, Lines 506-532:

"3.6 Discussion on the Limitations of the Algorithm

The current development of algorithms based on idealized steady-state CFD simulations relies on two key assumptions: low environmental turbulence intensity (0.1%) and turbulence length scales dominated by anemometer geometric parameters (0.012 meters). While this idealized setup effectively isolates rotor-induced flow distortion, its turbulence characteristics fundamentally differ from natural atmospheric conditions. However, it is crucial to emphasize that the algorithm's applicability under turbulent conditions remains valid. This is because rotor-induced wind speed deviations exhibit systemic long-time-scale characteristics, whereas atmospheric turbulence primarily affects measurement accuracy through random fluctuations in wind speed and direction with instantaneous nature. This temporal-scale distinction enables our

correction algorithm to effectively eliminate systemic biases while minimizing the impact of transient turbulence effects. Nevertheless, it should be noted that under stable atmospheric conditions (low wind speeds) as discussed in Sec. 3.5 or extreme weather scenarios, such airflow environments may disrupt the stable manoeuvrability of UAV rotors or obscure the systemic drainage effects of rotors, potentially leading to a nonlinear degradation in algorithm accuracy.

In addition, another limitation of our study is the assumption of a smooth surface in CFD simulations, which does not fully capture the impact of surface roughness on wind speed variations near the ground. In reality, surface roughness elements (e.g., vegetation, buildings, or terrain irregularities) alter the wind profile, increasing turbulence and wind shear in the atmospheric surface layer. This effect is particularly relevant for UAV-based wind measurements at low altitudes.

To further enhance the correction algorithm's applicability under diverse environmental conditions, future research will focus on the following aspects: conducting sensitivity studies under different turbulence intensity conditions, implementing supplementary correction modules specifically targeting atmospheric turbulence, and incorporating surface roughness length parameters in future CFD simulations. Although atmospheric turbulence presents significant challenges for UAVbased wind measurements, the correction framework established in this study has demonstrated its effectiveness in improving measurement accuracy across diverse meteorological conditions, thereby laying a critical foundation for developing reliable UAV-based wind measurement systems."

10. The validation results presented in Figure 9 reveal significant errors in wind speed and direction estimates, particularly under low wind conditions. A more comprehensive discussion of these errors would strengthen the manuscript by offering deeper insights into the algorithm's performance. For instance, exploring the correlation between VO, VR, and VT could provide valuable context, especially given the critical role of accurate wind fluctuation estimates in turbulence measurements. Furthermore, a clearer examination of the algorithm's limitations would greatly benefit the growing community of scientists employing UAV-based wind sensing algorithms, helping them better evaluate its potential impact and applicability.

**Response:** We thank the reviewer for their careful comment. This comment is similar to Comment 9, and we have revised the manuscript accordingly based on the previous feedback. These revisions are also applicable to the current comment.

---

## Author Comment (AC5)

Response to reviewer:

We greatly appreciate the reviewer's recognition of the value and significance of the present study, as well as the very valuable comments on the paper. We have addressed the comments carefully as detailed below. The original comments are in black italic and our replies in black normal font, we also put the revised paragraph in blue after each reply to show the changes.

*Suggestions for major revisions / questions:*

*1.Lines 67 to 100: While the main conclusion is correct, the authors avoid extracting uncertainty numbers for the measured wind states from the references.*

**Response:** We appreciate the reviewer's insightful comment. To address the concern regarding uncertainty quantification, we have revised Section 1 (Introduction) to explicitly incorporate uncertainty metrics from cited references, as detailed below:

[revised manuscript text omitted]

*2.Lines 135ff: The reviewer does not understand, why a matrix of ground-speed and wind-speed variations was used, instead of varying the airspeed? In other word: How is the groundspeed feed into the simulation?*

**Response:** We sincerely apologize for any confusion caused to the reviewers due to insufficient clarity in our original presentation. Regarding the wind speed parameter issue raised by the reviewer, please allow us to provide supplementary clarification:

During the initial design phase of the program, we adopted multi-parameter combinations of wind speed, wind direction, and ground speed to construct simulation scenarios. In the actual implementation of simulations, we converted wind speed and

ground speed into airspeed through vector synthesis for simulation calculations. This dual parametrization approach was primarily employed to ensure comprehensive coverage of the flight envelope in our simulation scenarios.

To enhance the clarity of presentation, we have added the following content in Section 2.2 (Lines 180-181) of the revised manuscript:

"It should be noted that the numerical simulations were conducted by converting wind speed and ground speed into airspeed through vector synthesis."

*3.Line 193, fig. 2, resp. ch. 2.4: The computational domain is described in detail, but I miss a reasoning for the chosen parameters.*

**Response:** We thank the reviewer for prompting us to clarify the rationale behind the setup of the computational domain.

In the revised manuscript, we have added the following explanation in blue text in Section 2.4 (Lines 213-215) to clarify the rationale for parameter selection:

"The simulation parameters primarily include the computational domain and mesh, fluid and environmental properties, as well as the rotating region. During the CFD flow simulations of the UAV using Solidworks, the computational domain dimensions (3.3 × 3.3 × 3.3 m³) were determined by prioritizing the analysis of flow field distribution around the anemometer while balancing computational costs."

*4.Line 382ff, ch. 3.5: The flight tests are not described adequately in quality and quantity. Thus the data basis of the uncertainty numbers given in line 399 – 401 is not clear. A link to the quite nice fig. 6 and 7 and the uncertainties is missing.*

**Response:** We sincerely appreciate the reviewer's constructive feedback. We would like to respectfully inquire whether Fig. 6 and 7 you referenced might actually correspond to Fig. 9 and 10. This is because Fig. 9 and 10 present the flight test data, whereas Fig. 6 and 7 focus on illustrating the algorithm construction process. If this is the case, to clarify the experimental design and strengthen the connection between flight tests, uncertainty quantification, and Fig. 9-10 (corresponding to Fig. 11-12 in the revised manuscript), we have made the following revisions in Lines 445-466 of Section 3.5 of the revised manuscript:

"A comparative experiment was designed to verify the effectiveness of the correction algorithm described in Eq. (23). The experiment primarily compares three different wind data: the first is the three-dimensional wind vector corrected only for UAV motion and attitude compensation (Eq. (19) and denoted as $V_O$), the second includes additional corrections for UAV rotor interference, along with motion and attitude compensation (Eq. (23) and denoted as $V_R$), and the third is the three-dimensional wind directly measured by the meteorological tower (denoted as $V_T$). The comparison experiment was conducted with the UAV flying wind-boxes around the 80-meter meteorological tower within the Experimental Base of the Beijing Key Laboratory of Cloud, Precipitation and Atmospheric Water Resources. The meteorological tower was equipped with three-dimensional ultrasonic anemometers positioned at heights of 30, 50, and 70 m, with one anemometer in the north and one in the south (see Fig. 10). Experiments were conducted during the daytime on July 19,

2022, with neutral atmospheric stability to minimize thermal boundary layer effects on vertical wind variability.

The UAV flew around the tower in a box flight path at a horizontal distance of about 10 m away from the tower, at all three heights. During these flights, the UAV maintained a commanded horizontal speed of approximately 5 m/s, a value selected as a compromise between achieving sufficient spatial sampling resolution and maintaining stable flight attitude control. A total of 30 independent wind-box flights were conducted, with each altitude (30, 50 m and 70 m) sampled 10 times. Each flight lasted approximately 13 minutes, generating over 800 valid data points per altitude. Given the potential interference from near-surface vegetation on the 30-meter anemometer on the tower, wind velocities acquired by the UAV at 50 and 70 m heights during steady flight intervals were analyzed herein. Using a 3σ threshold of the mean value of the entire dataset to exclude data outliers caused by sudden gusts or UAV maneuvers (such as turning), retaining data during steady UAV flight periods."

*Suggestion for minor revisions:*

*5.Line 249ff, fig. 3 to 5: I suggest a uniform scaling for the color bar.*

**Response:** We sincerely appreciate the reviewer's valuable suggestion regarding color bar consistency. We fully acknowledge the importance of maintaining consistency in visualization for effective comparison across figures.

We initially implemented uniform scaling (0-18 m/s) for Fig. 3-5 during revision. However, upon closer examination, we observed that the larger simulated airflow velocities in Fig. 3 and 5 fell within the yellow-green spectrum, making it challenging to visually distinguish subtle velocity variations critical to understanding rotor-induced airflow patterns.

After careful consideration, we opted to optimize the color bar ranges individually for each figure to enhance contrast in regions of scientific interest and maintain resolution of velocity gradients across different flight scenarios.

We would be happy to implement alternative visualization strategies if the reviewer feels this approach could be improved.

*6.Fig. 9: Instead of V0, VR, VT better write a full word description into the legend.*

**Response:** We would like to express our sincere gratitude to the reviewer's valuable suggestion. In the revised manuscript, we have added a complete description of the legend in Figure 11 (corresponding to Figure 9 in the original manuscript), as detailed below:

[Figure]